# Unsupervised Joint $k$-node Graph Representations with Compositional Energy-Based Models

**Leonardo Cotta**
Purdue University
cotta@purdue.edu

**Carlos H. C. Teixeira**
Universidade Federal de Minas Gerais, Brazil
carlos@dcc.ufmg.br

**Ananthram Swami**
United States Army Research Laboratory
ananthram.swami.civ@mail.mil

**Bruno Ribeiro**
Purdue University
ribeiro@cs.purdue.edu

## Abstract

Existing Graph Neural Network (GNN) methods that learn *inductive unsupervised* graph representations focus on learning node and edge representations by predicting observed edges in the graph. Although such approaches have shown advances in downstream node classification tasks, they are ineffective in jointly representing larger $k$-node sets, $k>2$. We propose MHM-GNN, an inductive unsupervised graph representation approach that combines joint $k$-node representations with energy-based models (hypergraph Markov networks) and GNNs. To address the intractability of the loss that arises from this combination, we endow our optimization with a loss upper bound using a finite-sample unbiased Markov Chain Monte Carlo estimator. Our experiments show that the unsupervised joint $k$-node representations of MHM-GNN produce better unsupervised representations than existing approaches from the literature.

## 1 Introduction

Inductive unsupervised learning using Graph Neural Networks (GNNs) in (dyadic) graphs is currently restricted to node and edge representations due to their reliance on edge-based losses [7, 15, 21, 46]. If we want to tackle downstream tasks that require jointly reasoning about $k > 2$ nodes, but whose input data are dyadic relations (i.e., standard graphs) rather than hyperedges, we must develop techniques that can go beyond edge-based losses.

Joint $k$-node representation tasks with dyadic relational inputs include drone swarms that communicate amongst themselves to jointly act on a task [41, 43], but also include more traditional product-recommendation tasks. For instance, an e-commerce website might want to predict which $k$ products could be jointly purchased in the same shopping cart, while the database only records (product, product) dyads to safeguard user information.

Srinivasan and Ribeiro [42] have recently shown that GNN node representations are insufficient to capture joint characteristics of $k$ nodes that are unique to this group of nodes. Indeed, our experiments show that using existing unsupervised GNN —with their node representations and edge losses— one cannot accurately detect these $k$-product carts on an e-commerce website. Unfortunately, existing GNN extensions that give joint $k$-node representations require *supervised* graph-wide losses [27, 25], leaving a significant gap between edge and *supervised* whole-graph losses (i.e., we need multiple labeled graphs for these to work). The main reason for this gap is scalability: to obtain *true unsupervised* joint $k$-node representations, one must optimize a model defined over *all* $k$-node induced subgraphs of a graph.

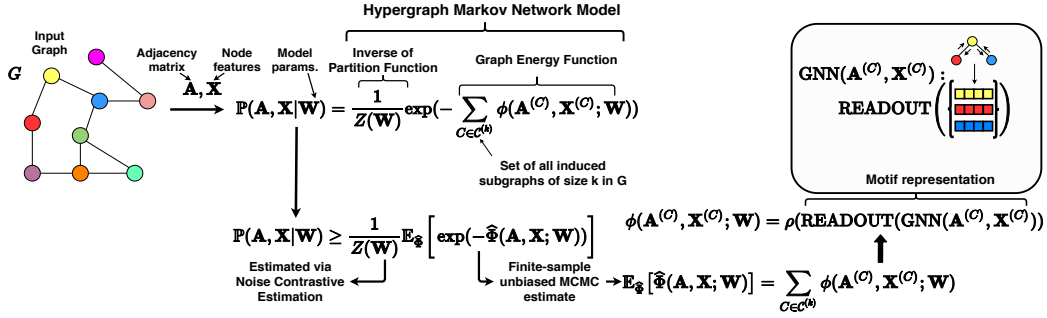

Figure 1: The proposed unsupervised graph representation using motif compositions. Here, we present the MHM-GNN model from Equation (1), the energy estimator $\widehat{\Phi}$ from Equation (4), the motif energy and representation from Equation (2).

Our approach MHM-GNN (Motif Hypergraph Markov Graph Neural Networks) leverages the compositionality of hypergraph Markov network models (HMNs) [38, 54, 22] that allows us to define an unsupervised objective (energy-based model) over GNN representations of motifs (see upper half of Figure 1).

Scalability is the main challenge we have to overcome, a type of scalability issue not addressed in the hypergraph Markov network literature [38, 54, 22]. First, there is the traditional likelihood intractability associated with computing the partition function $Z(\mathbf{W})$ of energy models —$Z(\mathbf{W})$ is shown in the likelihood $\mathbb{P}(\boldsymbol{A}, \boldsymbol{X}|\mathbf{W})$ in Figure 1 and also in Equation (1). There are standard solutions for this challenge (e.g., Noise-Contrastive Estimation (NCE) [14]). The more vexing challenge comes from the intractability created by our inductive graph representation that applies motif representations to all $k$-node subgraphs, which requires $\binom{n}{k}$ operations per gradient step, typically with $n \gg k$. To make this step tractable, we leverage recent advances in finite-sample unbiased Markov Chain Monte Carlo estimation for sums of subgraph functions over large graphs [45]. This unbiased estimate, combined with Jensen's inequality, allows us to optimize a lower bound on the intractable likelihood (assuming $Z(\mathbf{W})$ is known). Fold that into the asymptotics of NCE and we get a principled, tractable optimization.

**Contributions.** Our contributions are three-fold. First, we introduce MHM-GNN, which produces joint ($k > 2$)-node representations, where $k$ is a hyperparameter of the model. Second, we introduce a principled and scalable stochastic optimization method that learns MHM-GNN with a finite-sample unbiased estimator of the graph energy (see Fig. 1) and a NCE objective. Finally, we show how the joint $k$-node representations from MHM-GNN produce better unsupervised joint $k$-node representations than existing approaches that aggregate node representations. Our code is available at `https://github.com/PurdueMINDS/minds-mhm-gnn`.

## 2 Related Work

In this section, we briefly review existing approaches to *inductive unsupervised* representation learning of graphs, discuss existing work with higher-order graph representations and overview energy-based models. Finally, we present what in literature is not related to this work.

**Edge-based graph models.** Although graph models are prominent in many areas of research [29], most of the proposed models, such as the initial Erdös-Rényi model [10], stochastic block models [19] and the more recent neural network-based approaches [21, 15, 7] assume conditional independence of edges, resulting in what is often called an edge-based loss function. That is, all such models assume the appearance of edges in the graph is independent given the edge representations, which is usually computed via their endpoints' representations. This important conditional independence assumption appears in what we call edge-based graph models. There exist alternatives such as Markov Random Graphs [12], where an edge is dependent on every other edge that shares one of its endpoints, but graph models without any conditional independence assumption are still not commonly used.

**Inductive unsupervised node representations with GNNs.** Recently, GraphSAGE [15] introduced the use of GNNs to learn inductive node representations in an unsupervised manner by applying an edge-based loss while using short random walks. There are also auto-encoder approaches [21, 31, 39], where one tries to reconstruct the edges in the graph using node representations. Auto-encoders also

assume conditional independence of edges and can be classified as edge-based models. In contrast to edge-based loss models, DGI [46] minimizes the mutual entropy between node representations and a whole-graph representation —it does not model a probability distribution. Whilst the combination of GNNs and edge-based models has been shown to be effective in representing nodes and edges, *i.e.* $k = 1$ and $k = 2$ representations, moving to $k > 2$ joint representations requires a model with higher-order factorization. To this end, we introduce MHM-GNN, a model that leverages hypergraph Markov networks and GNNs to generate $k$-node motif representations.

**Joint $k$-node representations with dyadic graph.** Recently, Morris et al. [27] and Maron et al. [25] proposed higher-order neural architectures to represent entire graphs in a supervised learning setting, as opposed to the unsupervised setting discussed in this work. Moreover, we also point how since these higher-order GNN approaches are concerned with representing entire graphs in a supervised setting, the subgraph size $k$ is treated as a constant and scalability is not addressed. Our approach can incorporate higher-order GNNs and also the more recent Relational Pooling framework [28](see Equation (2)). We can summarize previous efforts to represent subgraphs in an unsupervised manner as sums of the individual nodes' representations [16]. Hypergraph neural network models [49, 2, 11] require observing polyadic data, while here we are interested in modeling dyadic data. We provide a broader discussion of higher-order graph models and the challenges of translating supervised approaches to an unsupervised setting in the supplement.

**Energy-based models.** Energy-Based Models (EBMs) have been widely used to learn representations of images [35], text [3], speech [44] and many other domains. In general, works in EBMs come in two flavors: what model to use for the energy and how to estimate the partition function $Z(\mathbf{W})$, which is usually intractable. For the latter, there are model-specific MCMC methods, such as Contrastive Divergence [17] and standard solutions, such as the one we choose in this work: Noise-Constrastive Estimation (NCE). As for the energy model, we opt for a hypergraph Markov network [38, 54, 22]. The energy of a graph is given by all of its $\binom{n}{k}$ subgraphs, which induces a new kind of intractability in the energy computation. Thus, we propose an unbiased energy estimation procedure in Section 4, which provides an upper bound on our NCE objective.

**Unrelated work.** It is important to not confuse *learning inductive unsupervised joint $k$-node* representations and other existing graph representation methods [27, 25, 13, 34, 36, 37, 23, 50]. Although motif-aware methods [23, 37] explicitly use motif information, they are used to build node representations rather than joint $k$-node representations, and thus, are equatable to other more powerful node representations, such as those in Hamilton et al. [15], Veličković et al. [46]. Here, we are interested in inductive tasks, hence transductive node representations, like Grover and Leskovec [13], Perozzi et al. [34], are unrelated. Nevertheless, as a matter of curiosity, we provide results for transductive node representations in our joint $k$-node tasks in the supplement, showing that our approach also works well compared to transductive settings even though our approach was not designed for transductive tasks. Supervised higher-order approaches [27, 25] extract whole-graph representations, which cannot be directly translated to existing unsupervised settings (see supplement for more details on these challenges). We are interested in methods that can be used in end-to-end representation learning, thus feature engineering and extraction, such as those used in graph kernels [50] are not of interest. Finally, the large body of work exploring hyperlink prediction in hypergraphs [6, 33, 53, 48, 51, 52] requires observing polyadic data (hypergraphs) and are transductive, as opposed to our work, where we consider observing dyadic data and propose an inductive model.

## 3 Motif Hypergraph Markov Graph Neural Networks (MHM-GNN)

In this section we start by introducing notation to then briefly introduce hypergraph Markov networks (HMNs), describe MHM-GNN with an HMN model, and discuss possible GNN-based energy functions used to represent motifs.

**Notation.** The $i$-th row of a matrix $\mathbf{M}$ will be denoted $\mathbf{M}_{i\cdot}$, and its $j$-th column $\mathbf{M}_{\cdot j}$. For the sake of simplicity, we will focus on graphs without edge attributes, even though our model can handle them using the GNN formulation from Battaglia et al. [5]. We denote a graph with $n$ nodes by $G = (V, E, \mathbf{X})$, where $V$ is the set of nodes, $E \subseteq V^2$ the edge set, $\mathbf{A} \in \{0, 1\}^{n \times n}$ its corresponding adjacency matrix and the matrix $\mathbf{X} \in \mathbb{R}^{n \times p}$ encodes the $p$ node features of all $n$ nodes. Each set of $k$ nodes from a graph $C \subseteq V : |C| = k$ has an associated induced subgraph

$G^{(C)} = (V^{(C)}, E^{(C)}, \boldsymbol{X}^{(C)})$ (see Definition 1). Induced subgraphs are also referred to as *motifs, graphlets, graph fragments or subgraphs*. Here, we will interchangeably refer to them as (induced) subgraphs or motifs.

**Definition 1** (Induced Subgraph). *Let $C \subseteq V : |C| = k$ be a set of $k$ nodes from $V$ with corresponding sorted sequence $\overrightarrow{C} = [C_1, ..., C_k] : C_i < C_{i+1}, C_i \in C \; \forall i \in \{1, ..., k\}$. Then, $G^{(C)} = (V^{(C)}, E^{(C)}, \boldsymbol{X}^{(C)})$ is the induced subgraph of $C$ in $G$, with adjacency matrix $\mathbf{A}^{(C)}$, where $V^{(C)} = \{1, ..., k\}$, $\boldsymbol{A} \in \{0,1\}^{k \times k} : \boldsymbol{A}_{ij}^{(C)} = \boldsymbol{A}_{C_i C_j}$ and $\boldsymbol{X}^{(C)} \in \mathbb{R}^{k \times p} : \boldsymbol{X}_{i.}^{(C)} = \boldsymbol{X}_{C_i \cdot}$.*

**Hypergraph Markov Networks (HMNs).** A Markov Network (MN) defines a joint probability distribution as a product of non-negative functions (potentials) over maximal cliques of an undirected graphical model [20, 4]. Although defined over maximal cliques, scalable techniques often assume factorization over non-maximal cliques [38, 4, 54], such as Pairwise Markov Networks (PMNs) [18], where the distribution is expressed as a product of edge potentials. In contrast, since we are interested in learning joint representations of $k$-node subgraphs, we need a hypergraph Markov network (HMN) (Definition 2), which is an MN model that can encompass all the variables of a $(k > 2)$-node subgraph.

Our graph model is an HMN. HMNs are to PMNs what hypergraphs are to graphs. In HMNs, the joint distribution is expressed as a product of potentials of hyperedges rather than edges. Since in HMNs potentials are defined over subsets of random variables of any size, we have the flexibility to do it over $k$-node subgraphs. There are previous works referring to HMNs as higher-order graphical models [38, 54], however we find the hypergraph analogy more clarifying. Next, we provide a formal definition of HMNs.

**Definition 2** (Hypergraph Markov Networks (HMNs)). *A hypergraph Markov network is a Markov network where the joint probability distribution of $\mathbf{Y} = \{Y_1, ..., Y_l\}$ can be expressed as $\mathbb{P}(\mathbf{Y} = \mathbf{y}) = \frac{1}{Z} \Pi_{h \in \mathcal{H}} \phi_h(\mathbf{y}_h)$, where $Z$ is the partition function $Z = \sum_{\mathbf{y}' \in \mathbf{Y}} \Pi_{h \in \mathcal{H}} \phi_h(\mathbf{y}'_h)$, $\phi_h(\cdot) \geq 0$ are non-negative, $\mathcal{H} \subseteq \mathcal{P}(\mathbf{Y}) \backslash \{\emptyset\}$, where $\mathcal{P}(\mathbf{Y})$ is the powerset of a set $\mathbf{Y}$, and $\mathcal{H}$ is the set of hyperedges in the Markov network, $\mathbf{Y}_h$ are the random variables associated with hyperedge $h$ and $\mathbf{y}, \mathbf{y}_h$ assignments of $\mathbf{Y}$ and $\mathbf{Y}_h$ respectively. Finally, an energy-based HMN assumes strictly positve potentials, resulting in the model $\mathbb{P}(\mathbf{Y} = \mathbf{y}) = \frac{1}{Z} \Pi_{h \in \mathcal{H}} \exp(-\phi_h(\mathbf{y}_h)) = \frac{1}{Z} \exp(-\sum_{h \in \mathcal{H}} \phi_h(\mathbf{y}_h))$, where $\phi_h(\cdot)$ is called the energy function of $h$.*

### 3.1 MHM-GNNs

We model $\mathbb{P}(\mathbf{A}, \boldsymbol{X} | \mathbf{W})$ with an energy-based HMN, as described in Definition 2, where a hyperedge corresponds to an induced subgraph of $k$ nodes in the graph $G$. More precisely, for every set of $k > 1$ nodes in the graph $C \subseteq V, |C| = k$, we define a hyperedge $h = \{\mathbf{A}_{ij} : (i,j) \in C^2\} \cup \{\boldsymbol{X}_{i,\cdot} : i \in C\}$ in the HMN to encompass every node variable in the $k$-node set and every edge variable with both endpoints in it. A hyperedge can be indexed by a set of nodes $C$, since its corresponding set of random variables is given by the features $\boldsymbol{X}^{(C)}$ and the adjacency matrix $\mathbf{A}^{(C)}$ of the subgraph induced by $C$, following Definition 1. Thus, a graph with $n$ nodes will have an HMN with $\binom{n}{k}$ potentials. We formally define the model in Definition 3.

**Definition 3** (MHM-GNN). *Let $\mathcal{C}^{(k)}$ denote the set of all $\binom{n}{k}$ combinations of $k$ nodes from $G$. We define a hypergraph Markov Network with a set of hyperedges $\{\{\boldsymbol{A}_{ij} : (i,j) \in C\} \cup \{\boldsymbol{X}_{i,\cdot} : i \in C\} : C \in \mathcal{C}^{(k)}\}$, which following Definitions 1 and 2, entails the model*

$$\mathbb{P}(\mathbf{A}, \boldsymbol{X} | \mathbf{W}) = \frac{\exp\left(-\sum_{C \in \mathcal{C}^{(k)}} \phi(\mathbf{A}^{(C)}, \boldsymbol{X}^{(C)}; \mathbf{W})\right)}{Z(\mathbf{W})}, \tag{1}$$

*where $\phi(\cdot, \cdot; \mathbf{W})$ is an energy function with parameters $\mathbf{W}$ and $Z(\mathbf{W})$ is the partition function given by $Z(\mathbf{W}) = \sum_{n=1}^{\infty} \sum_{\mathbf{A}' \in \{0,1\}^{n \times n}} \int_{\boldsymbol{X}' \in \mathbb{R}^{n \times p}} \exp(-\sum_{C \in \mathcal{C}^{(k)}} \phi(\mathbf{A}'^{(C)}, \boldsymbol{X}'^{(C)}; \mathbf{W})) d\boldsymbol{X}'$.*

Although MHM-GNN factorizes the total energy of a graph, the model does not assume any conditional independence between edge variables for $k > 3$. For $k = 2$, the model recovers existing edge-based models and for $k = 3$ edge variables are dependent only on edges that share one of their endpoints, recovering the Markov random graphs class [12]. Furthermore, MHM-GNN will learn a jointly exchangeable distribution [30] if the subgraph energy function $\phi(., .; \mathbf{W})$ is jointly exchangeable, such as a GNN. In the supplement we connect MHM-GNN assumptions, exchangeability and Exponential Random Graph Models (ERGMs).

**Subgraph energy function and representations.** As mentioned, to have a jointly exchangeable model with MHM-GNN, we need an energy function $\phi(\mathbf{A}^{(C)}, \boldsymbol{X}^{(C)}; \mathbf{W})$ that is jointly exchangeable with respect to the subgraph $G^{(C)}$. To this end, we break down $\phi(\mathbf{A}^{(C)}, \boldsymbol{X}^{(C)}; \mathbf{W})$ into a composition of two functions. First, we compute a jointly exchangeable representation of $G^{(C)}$, then we use it as input to a more general function that assigns an energy value to the subgraph. Following recent GNN advances [9, 47, 27], we define the subgraph representation with a permutation invariant (READOUT) function over the nodes' representations given by a GNN, denoted by $h^{(C)}(\mathbf{A}^{(C)}, \boldsymbol{X}^{(C)}; \mathbf{W}_{\text{GNN}}, \mathbf{W}_{\text{R}}) = \text{READOUT}(\text{GNN}(\mathbf{A}^{(C)}, \boldsymbol{X}^{(C)}; \mathbf{W}_{\text{GNN}}); \mathbf{W}_{\text{R}})$. Usually, the READOUT function is a row-wise sum followed by a multi-layer perceptron. Note that, although we choose a 1-GNN approach to represent the subgraph here, any jointly exchangeable graph representation can be used to represent the subgraph, such as $k$-GNNs [27] and Relational Pooling [28].

Finally, we can define the energy of a subgraph $G^{(C)}$ as

$$\phi(\mathbf{A}^{(C)}, \boldsymbol{X}^{(C)}; \mathbf{W}) = \mathbf{W}_{\text{energy}}^T \rho(h^{(C)}(\mathbf{A}^{(C)}, \boldsymbol{X}^{(C)}; \mathbf{W}_{\text{GNN}}, \mathbf{W}_{\text{R}}); \mathbf{W}_\rho) \tag{2}$$

where the model set of weights is $\mathbf{W} = \{\mathbf{W}_{\text{energy}}, \mathbf{W}_{\text{R}}, \mathbf{W}_\rho, \mathbf{W}_{\text{GNN}}\}$, $\rho(\cdot; \mathbf{W}_\rho)$ is a permutation sensitive function with parameters $\mathbf{W}_\rho$ such as a multi-layer perceptron with range in $\mathbb{R}^{1 \times H}$ and $\mathbf{W}_{\text{energy}} \in \mathbb{R}^{1 \times H}$ is a (learnable) weight matrix.

Although the functional form of the distribution and subgraph representations are properly defined, directly computing both the partition function and the total energy of a graph are computationally intractable for an arbitrary $k$. Therefore, in the next section we discuss how to properly learn the distribution parameters, providing a principled and scalable approximate method.

## 4 Learning MHM-GNNs

In this section, we first define our unsupervised objective through Noise-Contrastive Estimation (NCE) and then show how to approximate it.

**Noise-Contrastive Estimation (NCE).** Since directly computing $Z(\mathbf{W})$ of Equation (1) is intractable, we use Noise-Contrastive Estimation (NCE) [14]. In NCE, the model parameters are learned by contrasting observed data and negative (noise) sampled examples. Given the set $\mathcal{D}_{\text{true}}$ of observed graphs and $M|\mathcal{D}_{\text{true}}|$ sampled noise graphs from a noise distribution $\mathbb{P}_n(\mathbf{A}, \boldsymbol{X})$ composing the set $D_{\text{noise}}$, we can define the loss function to be minimized as

$$\mathcal{L}(\mathbf{A}, \boldsymbol{X}; \mathbf{W}) = - \sum_{\mathbf{A} \in \mathcal{D}_{\text{true}}} \log(\hat{y}(\Phi(\mathbf{A}, \boldsymbol{X}; \mathbf{W}), \mathbb{P}_n(\mathbf{A}, \boldsymbol{X})))$$
$$- \sum_{\mathbf{A} \in \mathcal{D}_{\text{noise}}} \log(1 - \hat{y}(\Phi(\mathbf{A}, \boldsymbol{X}; \mathbf{W}), \mathbb{P}_n(\mathbf{A}, \boldsymbol{X}))).$$

with $\hat{y}(\Phi(\mathbf{A}, \boldsymbol{X}; \mathbf{W}), \mathbb{P}_n(\mathbf{A}, \boldsymbol{X})) = \sigma(-\Phi(\mathbf{A}, \boldsymbol{X}; \mathbf{W}) - \log(M\mathbb{P}_n(\mathbf{A}, \boldsymbol{X})))$, where $\sigma(\cdot)$ is the sigmoid function and $\Phi(\mathbf{A}, \boldsymbol{X}; \mathbf{W}) = \sum_{C \in \mathcal{C}^{(k)}} \phi(\mathbf{A}^{(C)}, \boldsymbol{X}^{(C)}; \mathbf{W})$ denotes the total energy of a graph $G = (V, E, \boldsymbol{X})$ in MHM-GNN.

If the largest graph in $\mathcal{D}_{\text{true}} \cup \mathcal{D}_{\text{noise}}$ has $n$ nodes, directly computing the gradient of the loss $\nabla \mathcal{L}(\mathbf{A}, \boldsymbol{X}; \mathbf{W})$ would take $\mathcal{O}(M|\mathcal{D}_{\text{true}}|^2 n^k)$ operations. Traditional Stochastic Gradient Descent (SGD) methods get rid of the dataset size $M|\mathcal{D}_{\text{true}}|^2$ term by uniformly sampling graph examples. Thus, naively optimizing the NCE loss with SGD would still require $\mathcal{O}(n^k)$ operations to compute $\Phi(\mathbf{A}, \boldsymbol{X}; \mathbf{W})$. In what follows we rely on a stochastic optimization procedure that requires a finite-sample unbiased estimator of $\Phi(\mathbf{A}, \boldsymbol{X}; \mathbf{W})$, where we can also control the estimator's variance with a hyperparameter. We show that the resulting stochastic optimization is theoretically sound by proving that it optimizes an upper bound of the original loss.

**Estimating the MHM-GNN energy** $\Phi(\mathbf{A}, \boldsymbol{X}; \mathbf{W})$**.** To estimate $\Phi(\mathbf{A}, \boldsymbol{X}; \mathbf{W})$, we need to first observe that —due to sparsity in real-world graphs— an arbitrary set of $k$ nodes from a graph will induce an empty subgraph with high probability [29]. Therefore, to estimate $\Phi(\mathbf{A}, \boldsymbol{X}; \mathbf{W})$ with low variance, we focus on estimating it on *connected induced subgraphs* (CISes) [45], while assuming some constant high energy for disconnected subgraphs. To this end, if $\mathcal{C}_{\text{conn}}^{(k)}$ is the set of all $k$-node

sets that induce a connected subgraph in $G$, we are now making the reasonable assumption

$$\Phi(\mathbf{A}, \mathbf{X}; \mathbf{W}) = \sum_{C \in \mathcal{C}_{\text{conn}}^{(k)}} \phi(\mathbf{A}^{(C)}, \mathbf{X}^{(C)}; \mathbf{W}) + \text{constant}, \qquad (3)$$

where w.l.o.g. we assume the constant to be zero. Since enumerating all CISes is computationally intractable for arbitrary $k$ [8], we introduce next a finite-sample unbiased estimator for $\Phi(\mathbf{A}, \mathbf{X}; \mathbf{W})$ of Equation (3) over CISes, denoted by $\widehat{\Phi}(\mathbf{A}, \mathbf{X}; \mathbf{W})$.

We start by presenting the concept of the higher-order network ($k$-HON) of a graph $G$ and its variant called *collapsed node* HON ($k$-CNHON). An ordinary $k$-HON $G^{(k)}$ is a network where the nodes $V^{(k)}$ correspond to $k$-node CISes from $G$ and edges $E^{(k)}$ connect two CISes that share $k-1$ nodes. On the other hand, a $k$-CNHON or $G^{(k,\mathcal{I})}$ is a multigraph where a subset of the nodes of $G^{(k)}$, $\mathcal{I} \subset V^{(k)}$, are collapsed into a single node in $G^{(k,\mathcal{I})}$. The collapsed node, henceforth denoted *the supernode*, is now node $v_{\mathcal{I}}^{(k)}$ in $G^{(k,\mathcal{I})}$. The edges in $G^{(k)}$ of the collapsed nodes $v \in \mathcal{I}$ among themselves, i.e., the edges in $\mathcal{I} \times \mathcal{I}$, do not exist in $G^{(k,\mathcal{I})}$. The edges between the collapsed nodes $v \in \mathcal{I}$ and other nodes $V \setminus \mathcal{I}$ are added to $G^{(k,\mathcal{I})}$ by replacing the endpoint $v$ with endpoint $v_{\mathcal{I}}^{(k)}$, making $G^{(k,\mathcal{I})}$ a multigraph (a graph with multiple edges between the same two nodes). All the remaining edges in $G^{(k)}$ are preserved in $G^{(k,\mathcal{I})}$. In Figure 2 we show a graph and its $k$-CNHON with a Random Walk Tour (Definition 4) example. A formal definition is given in supplement.

**Definition 4** (Random Walk Tour (RWT))**.** *Consider a simple random walk over a multigraph starting at node $v_{init}$. A Random Walk Tour (RWT) is represented by a sequence of nodes $\mathcal{T} = \{v_1, ..., v_t, v_{t+1}\}$ visited by the random walk such that $v_1 = v_{init}$, $v_{t+1} = v_{init}$ and $v_i \neq v_{init} \; \forall \; 1 < i < t+1$.*

In this work, we construct the estimator $\widehat{\Phi}(\mathbf{A}, \mathbf{X}; \mathbf{W})$ via *random walk tours* (RWTs) on the $k$-CNHON $G^{(k,\mathcal{I})}$ starting at the collapsed node $v_{\mathcal{I}}^{(k)}$ (i.e, $v_{\text{init}} = v_{\mathcal{I}}^{(k)}$ in Definition 4). As previously introduced and discussed in Avrachenkov et al. [1] and Teixeira et al. [45], increasing the number of tours and the supernode size allow for variance reduction. Using these insights, we propose the estimator $\widehat{\Phi}(\mathbf{A}, \mathbf{X}; \mathbf{W})$, whose properties are defined in Theorem 1.

**Theorem 1.** *Let $G^{(k)}$ be the $k$-HON of a graph $G$, a set $\mathcal{I}$ of $k$-node sets that induce CISes in $G$ (as described above) and $N^{(k)}(C)$ the set of neighbors of the corresponding node of CIS $C$ in $G^{(k)}$. In addition, consider the sample-path $\mathcal{T}^r = (v_1^r, ..., v_{t^r}^r, v_{t^r+1}^r)$ visited by the $r$-th RWT on $G^{(k,\mathcal{I})}$ starting from supernode $v_{\mathcal{I}}^{(k)}$, where $v_i^r$ is the node reached at step $i$ for $1 \leq r \leq q$ (Definition 4), and $q \geq 1$ is the number of RWTs. Since $\mathcal{T}^r$ is a RWT, $v_1^r = v_{\mathcal{I}}^{(k)}$, $v_{t^r+1}^r = v_{\mathcal{I}}^{(k)}$ and $v_i^r \neq v_{\mathcal{I}}^{(k)} : 1 < i < t^r + 1$. The nodes $(v_2^r, ..., v_{t^r}^r)$ in the sample path $\mathcal{T}^r$ have a corresponding sequence of induced $k$-node subgraphs in the graph $G$, denoted $\mathcal{T}_C^r = (C_i^r)_{i=2}^{t^r}$. Then, the estimator*

$$\widehat{\Phi}(\mathbf{A}, \mathbf{X}; \mathbf{W}) = \underbrace{\sum_{v \in \mathcal{I}} \phi(\mathbf{A}^{(v)}, \mathbf{X}^{(v)}; \mathbf{W})}_{\text{Energy of } k\text{-node CISes in } \mathcal{I} \text{ (supernode)}} + \underbrace{\left( \frac{\sum_{u \in \mathcal{I}} |N^{(k)}(u) \setminus \mathcal{I}|}{q} \right) \sum_{r=1}^{q} \sum_{i=2}^{t^r} \frac{\phi(\mathbf{A}^{(C_i^r)}, \mathbf{X}^{(C_i^r)}; \mathbf{W})}{|N^{(k)}(C_i^r)|}}_{\text{RWT-estimated energy of remaining } k\text{-node CISes in } G}$$

*is an unbiased and consistent estimator of $\Phi(\mathbf{A}, \mathbf{X}; \mathbf{W})$ in Equation (3) with constant=0.*

The proof of Theorem 1 is in the supplement.

We can now replace $\Phi(\mathbf{A}, \mathbf{X}; \mathbf{W})$ in $\mathcal{L}(\mathbf{A}, \mathbf{X}; \mathbf{W})$ with its estimator $\widehat{\Phi}(\mathbf{A}, \mathbf{X}; \mathbf{W})$, resulting in a loss estimate $\widehat{\mathcal{L}}(\mathbf{A}, \mathbf{X}; \mathbf{W})$. It follows from Theorem 1 and Jensen's inequality that our loss estimate is in expectation an upper bound to the true NCE loss, *i.e.* $\mathbb{E}_{\widehat{\Phi}}[\widehat{\mathcal{L}}(\mathbf{A}, \mathbf{X}; \mathbf{W})] \geq \mathcal{L}(\mathbf{A}, \mathbf{X}; \mathbf{W})$. Moreover, note that using an estimator of this nature in higher-order GNNs, such as $k$-GNNs [27], does not allow for a bound in the loss estimation (please, see the supplement for further discussion). Note that the variance of $\widehat{\Phi}$ is controlled by the hyperparameter $q$, the number of tours.

## 5 Results

In this section, we evaluate the quality of the unsupervised motif representations learned by MHM-GNN over six datasets using two joint $k$-node transfer learning tasks. The tasks consider three citation

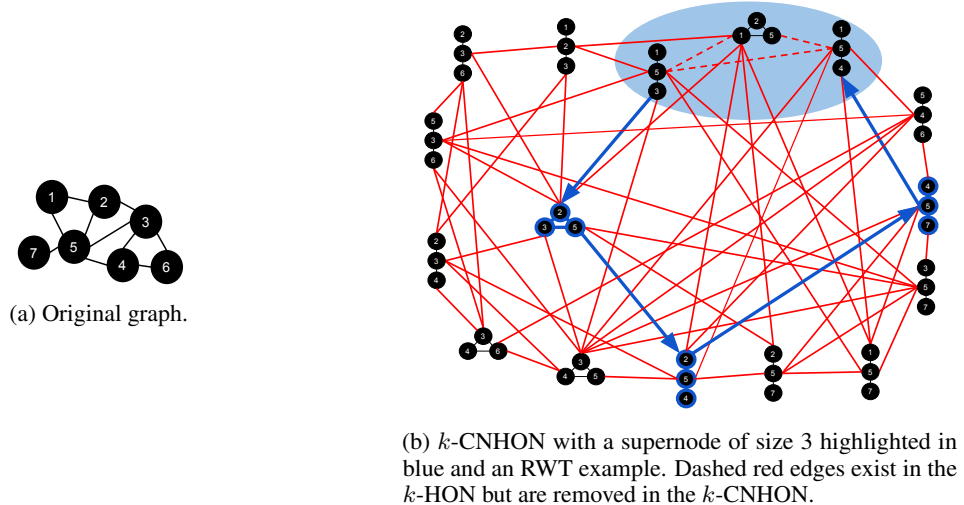

(a) Original graph.

(b) $k$-CNHON with a supernode of size 3 highlighted in blue and an RWT example. Dashed red edges exist in the $k$-HON but are removed in the $k$-CNHON.

Figure 2: A graph and its corresponding $k$-CNHON with an RWT example.

networks, one coauthorship network and two product networks to show how the pre-trained motif representations consistently outperform pooling pre-trained node representations in predicting $k$-node hidden hyperedge labels in downstream tasks — details of these tasks are in the *Hyperedge Detection* and *DAG Leaf Counting* subsections.

A good $k$-node representation of a graph is able to capture hidden $k$-order relationships while only observing pairwise interactions. To this end, our tasks evaluate the quality of the unsupervised representations using two hidden hyperedge label prediction tasks. Using the pre-trained unsupervised learned representation as input, we train a simple logistic regression classifier to predict the hidden hyperedge label of a $k$-node set.

**Datasets.** We use the Cora, Citeseer and Pubmed [40] citation networks, the DBLP coauthorship network [49], the Steam [32] and the Rent the Runway [26] product networks (more details about the datasets are in the supplement). These datasets were chosen since they contain joint $k$-node information. In the coauthorship network, nodes correspond to authors and edges to the coauthorship of a paper, hidden from the training data we also have the papers and their corresponding author list. In the product networks, nodes correspond to products and an edge exists if the same user bought the two end-point products, hidden from the training data we have the list of products each user bought. In the citation network, nodes correspond to papers and edges to citations, hidden from the training data we also have the direction in which the citation occurred. These directions, paper author list and users purchase history which are hidden in the training data used by the unsupervised GNN and MHM-GNN representations, give us two transfer learning *k-node downstream tasks*, described in what follows.

**Hyperedge Detection.** This hyperedge task, inspired by Yadati et al. [49], creates a $k$-node hyperedge in a citation network whenever a paper cites $k - 1$ other papers, in a coauthorship network whenever $k$ authors write a paper together and in a product network whenever a user buys $k$ products. Examples are in the citation networks $k$-size subgraphs with at least one node with degree $k - 1$ and $k$-cliques in the other networks. Note how Yadati et al. [49] directly learns its representations from the hypergraph, a significantly easier task. The downstream classifier —a simple logistic regression classifier— uses the unsupervised pre-trained representations to classify whether a set of $k$ nodes forms a (hidden) hyperege or not. This task allows us to compare the quality of the unsupervised node representations of GNNs against that of MHM-GNN.

**DAG Leaf Counting.** This task considers the citation networks. Again, baselines and MHM-GNN are trained over the undirected graphs. Due to the temporal order of citations, subgraphs correspond to Directed Acyclic Graphs (DAGs) in the directed structure. For a connected $k$-node induced subgraph in the directed graph, we want to predict the number of leaves of the resulting DAG. Again, the downstream classifier —a simple logistic regression classifier— uses the unsupervised pre-trained representations of a set of $k$-nodes to predict the exact number of leaves formed by the (hidden) $k$-node DAG. The number of leaves defines the number of influential papers in the $k$-node set.

**MHM-GNN architecture.** The energy function of MHM-GNN is as described in Equation (2), where we use a one-hidden layer feedforward network with LeakyReLU activations as $\rho$, a row-wise sum followed by also a one-hidden layer feedforward network with LeakyReLU activations as the READOUT function and a single layer GraphSAGE-mean Hamilton et al. [15] as the GNN.

**Training the model.** Since the datasets used in this section contain only one large graph for training —as in most of the real-world graph datasets— we need to construct a larger set of positive examples $\mathcal{D}_{\text{true}}$ to learn the distribution $\mathbb{P}(\mathbf{A}, \boldsymbol{X}|\mathbf{W})$. One way to overcome this issue is by subsampling the original large graph. While sampling smaller graphs that preserve the original graph properties, we can approximate the true $\mathbb{P}(\mathbf{A}, \boldsymbol{X}|\mathbf{W})$ distribution and control the complexity of $\widehat{\Phi}(\mathbf{A}, X; \mathbf{W}))$ (since tour return times are affected by the size of the graph). To this end, we construct $\mathcal{D}_{\text{true}}$ by subsampling the original graph with Forest Fire [24]. As for the noise distribution, we turn to the one used by Veličković et al. [46], where for each positive example we generate $M$ negative samples by keeping the adjacency matrix and shuffling the feature matrix. This noise distribution allows us to keep structural properties of the graph, *e.g.* connectivity, while significantly changing how node features affect the distribution. We precisely describe all hyperparameters and hyperparameter tuning in the supplement.

**Experimental setup.** To evaluate the performance of the pre-trained MHM-GNN representations in the above downstream tasks, we first train the model accordingly for $k = 3$ and $k = 4$ motif sizes over all six datasets. In the citation and coauthorship networks, we have a single graph, thus these tasks require dividing the graph into training and test sets when evaluating the representations, such that the distribution of observed subgraphs is preserved. To this end, for each dataset, we perform min-cut clustering and use the two cuts for training and test data in the downstream task. For the product networks, to explore the inductive nature of our method, we create two graphs, one for training the models and one for testing the representations. For the Steam dataset, we train on the user-product data from 2014 and test considering the data from 2015. Similarly, for the Rent the Runway dataset, we train on data from 2016 and test on data from 2017. Tables 1 and 2 show our results for the Hyperedge Detection and Table 3 for the DAG Leaf Counting tasks. MHM-GNN uses motif sizes $k = 3, 4$. In the supplement, we also show results for $k = 5$. For each task (and $k$), we report the mean and the standard deviation of the balanced accuracy (mean recall of each class) achieved by logistic regression over five different runs. Furthermore, the pre-trained representations (baselines and our approach) have dimension 128. Additional implementation details and hyperparameter search can be found in the supplement.

**Baselines.** We evaluate the motif representations from MHM-GNN against two alternatives representing the $k$ nodes using state-of-the-art unsupervised GNN representations: GraphSAGE [15] and Deep Graph Infomax (DGI) [46]. As a naive baseline, we compare against summing the original features from the nodes, *i.e.* a representation that ignores structural information. Moreover, we compare our pre-trained MHM-GNN representations with an untrained (random parameters) version of MHM-GNN. Further, since the citation and coauthorship networks consider single graphs, in the supplement we show results for these datasets with two prominent transductive node embedding methods [34, 13], evidencing how even in transductive settings node embeddings fail to capture joint $k$-node relationships.

**Results.** The hidden hyperedge downstream tasks are designed to better understand how well pre-trained unsupervised representations can capture joint $k$-node properties. A good joint $k$-node representation should be able to disentangle (hidden) polyadic relationships, even though they only have access to dyadic data. In our Hyperedge Detection task using pre-trained unsupervised representations, Tables 1 and 2 show that MHM-GNN representations consistently outperform GNN node representations across all datasets. In particular, MHM-GNN increases classification accuracy by up to 11% over the best-performing baseline. The results of our the DAG Leaf Counting task, shown in Table 3, reinforce that pre-trained unsupervised MHM-GNN representations can better capture joint $k$-node interactions. In particular, MHM-GNN representations observe classification accuracies by up to 24% in this downstream task.

*Node representations and joint $k$-node graph tasks.* Our experiments further validate the theoretical claims in Srinivasan and Ribeiro [42], that structural node representations are not capable of performing joint $k$-node tasks. That is, the inductive node representations baselines perform similarly to a random classifier in most settings in Tables 1 to 3. In contrast, the greater accuracy of MHM-GNN shows that joint $k$-node representations are informative.

Table 1: Balanced accuracy for the **Hyperedge Detection** task over subgraphs of size $k = 3$. We report mean and standard deviation over five runs.

| Method | Cora $k = 3$ | Citeseer $k = 3$ | Pubmed $k = 3$ | DBLP $k = 3$ | Steam $k = 3$ | Rent the Runway $k = 3$ |
|---|---|---|---|---|---|---|
| GS-mean[15] | $0.490 \pm 0.03$ | $0.509 \pm 0.07$ | $0.499 \pm 0.00$ | $0.560 \pm 0.08$ | $0.565 \pm 0.01$ | $0.665 \pm 0.00$ |
| GS-max[15] | $0.486 \pm 0.04$ | $0.493 \pm 0.06$ | $0.498 \pm 0.00$ | $0.569 \pm 0.06$ | $0.579 \pm 0.02$ | $0.667 \pm 0.00$ |
| GS-lstm[15] | $0.483 \pm 0.04$ | $0.486 \pm 0.05$ | $0.510 \pm 0.02$ | $0.585 \pm 0.06$ | $0.518 \pm 0.01$ | $0.518 \pm 0.01$ |
| DGI[46] | $0.487 \pm 0.03$ | $0.508 \pm 0.07$ | $0.509 \pm 0.02$ | $0.497 \pm 0.00$ | $0.588 \pm 0.01$ | $0.612 \pm 0.00$ |
| Raw Features | $0.499 \pm 0.00$ | $0.588 \pm 0.00$ | $0.502 \pm 0.00$ | $0.518 \pm 0.00$ | $0.534 \pm 0.00$ | $0.649 \pm 0.00$ |
| MHM-GNN (Rnd) | $0.498 \pm 0.00$ | $0.520 \pm 0.05$ | $0.498 \pm 0.01$ | $0.491 \pm 0.01$ | $0.571 \pm 0.01$ | $0.650 \pm 0.00$ |
| MHM-GNN | $\mathbf{0.618} \pm 0.03$ | $\mathbf{0.621} \pm 0.01$ | $\mathbf{0.602} \pm 0.06$ | $\mathbf{0.773} \pm 0.02$ | $\mathbf{0.611} \pm 0.01$ | $\mathbf{0.676} \pm 0.00$ |

Table 2: Balanced accuracy for the **Hyperedge Detection** task over subgraphs of size $k = 4$. We report mean and standard deviation over five runs.

| Method | Cora $k = 4$ | Citeseer $k = 4$ | Pubmed $k = 4$ | DBLP $k = 4$ | Steam $k = 4$ | Rent the Runway $k = 4$ |
|---|---|---|---|---|---|---|
| GS-mean[15] | $0.450 \pm 0.11$ | $0.544 \pm 0.03$ | $0.524 \pm 0.05$ | $0.530 \pm 0.15$ | $0.640 \pm 0.03$ | $0.851 \pm 0.00$ |
| GS-max[15] | $0.462 \pm 0.09$ | $0.538 \pm 0.04$ | $0.558 \pm 0.05$ | $0.511 \pm 0.14$ | $0.688 \pm 0.01$ | $0.855 \pm 0.00$ |
| GS-lstm[15] | $0.444 \pm 0.09$ | $0.536 \pm 0.04$ | $0.566 \pm 0.06$ | $0.653 \pm 0.02$ | $0.504 \pm 0.01$ | $0.546 \pm 0.03$ |
| DGI[46] | $0.463 \pm 0.10$ | $0.526 \pm 0.04$ | $0.549 \pm 0.06$ | $0.500 \pm 0.00$ | $0.664 \pm 0.02$ | $0.749 \pm 0.03$ |
| Raw Features | $0.529 \pm 0.01$ | $0.581 \pm 0.00$ | $0.498 \pm 0.02$ | $0.558 \pm 0.01$ | $0.535 \pm 0.01$ | $0.857 \pm 0.00$ |
| MHM-GNN (Rnd) | $0.490 \pm 0.10$ | $0.478 \pm 0.03$ | $0.510 \pm 0.02$ | $0.492 \pm 0.02$ | $0.679 \pm 0.01$ | $0.832 \pm 0.01$ |
| MHM-GNN | $\mathbf{0.575} \pm 0.03$ | $\mathbf{0.659} \pm 0.08$ | $\mathbf{0.701} \pm 0.10$ | $\mathbf{0.740} \pm 0.05$ | $\mathbf{0.750} \pm 0.00$ | $\mathbf{0.860} \pm 0.00$ |

*Ablation study.* As an ablation, we test whether our optimization in MHM-GNN improves the unsupervised joint $k$-node representations, when compared against random neural network weights. And while Tables 1 to 3 show that MHM-GNN with random weights perform well in the tasks, since they are effectively a type of motif feature, the higher accuracy of the optimized joint $k$-node representations shows that the optimized representations in MHM-GNN are indeed learned.

Table 3: Balanced accuracy for the **DAG Leaf Counting** task over subgraphs of size $k = 3$ and $k = 4$. We report mean and standard deviation over five runs.

| Method | Cora $k = 3$ | Cora $k = 4$ | Citeseer $k = 3$ | Citeseer $k = 4$ | Pubmed $k = 3$ | Pubmed $k = 4$ |
|---|---|---|---|---|---|---|
| GS-mean[15] | $0.468 \pm 0.05$ | $0.245 \pm 0.06$ | $0.492 \pm 0.04$ | $0.356 \pm 0.02$ | $0.502 \pm 0.00$ | $0.384 \pm 0.03$ |
| GS-max[15] | $0.467 \pm 0.06$ | $0.245 \pm 0.07$ | $0.486 \pm 0.04$ | $0.347 \pm 0.01$ | $0.499 \pm 0.00$ | $0.371 \pm 0.03$ |
| GS-lstm[15] | $0.473 \pm 0.05$ | $0.263 \pm 0.07$ | $0.482 \pm 0.04$ | $0.348 \pm 0.01$ | $0.507 \pm 0.02$ | $0.372 \pm 0.03$ |
| DGI[46] | $0.478 \pm 0.05$ | $0.278 \pm 0.07$ | $0.504 \pm 0.06$ | $0.350 \pm 0.02$ | $0.505 \pm 0.02$ | $0.362 \pm 0.03$ |
| Raw Features | $0.501 \pm 0.01$ | $0.325 \pm 0.00$ | $0.567 \pm 0.00$ | $0.380 \pm 0.00$ | $0.503 \pm 0.00$ | $0.339 \pm 0.00$ |
| MHM-GNN (Rnd) | $0.497 \pm 0.00$ | $0.327 \pm 0.00$ | $0.518 \pm 0.04$ | $0.319 \pm 0.01$ | $0.499 \pm 0.01$ | $0.343 \pm 0.01$ |
| MHM-GNN | $\mathbf{0.593} \pm 0.03$ | $\mathbf{0.452} \pm 0.03$ | $\mathbf{0.606} \pm 0.01$ | $\mathbf{0.469} \pm 0.02$ | $\mathbf{0.626} \pm 0.02$ | $\mathbf{0.475} \pm 0.08$ |

As opposed to node GNN representations and other non-compositional unsupervised graph representation approaches, MHM-GNN does not take graph-wide information as input. Thus, it is natural to wonder to what extent pre-trained MHM-GNN joint $k$-node representations are informative of the entire graph to which they belong to. Hence, in the supplement we consider whole-graph classification as the downstream task. In this setting, we show how composing (by pooling) MHM-GNN motif representations can perform better than non-compositional methods, further indicating how our learned motif representations can capture the underlying graph distribution $\mathbb{P}(\mathbf{A}, \boldsymbol{X}; \mathbf{W})$.

# 6   Conclusions

By combining hypergraph Markov networks, an unbiased finite-sample MCMC estimator, and graph representation learning, we introduced MHM-GNN, a new scalable class of energy-based representation learning methods capable of learning joint $k$-node representations over dyadic graphs in an *inductive unsupervised* manner. Finally, we show how pre-trained MHM-GNN representations achieve more accurate results in downstream joint $k$-node tasks. The energy-based optimization in this work allows for many extensions, such as designing different $k$-node subgraph representation learning methods, new subgraph function estimators for MHM-GNN's loss function, and formulating new joint $k$-node tasks.

## Broader Impact

This work presents an unsupervised model together with a stochastic optimization procedure to generate $k$-node representations from graphs, such as online social networks, product networks, citation networks, coauthorship networks, etc. As is the case with any learning algorithm, it is susceptible to produce biased representations if trained with biased data. Moreover, although the representations might be bias free, the downstream task defined by the user might be biased and thus, also produce biased decisions.

## Acknowledgments

This work was funded in part by the National Science Foundation (NSF) Awards CAREER IIS-1943364, CCF-1918483, and by the ARO, under the U.S. Army Research Laboratory contract number W911NF-09-2-0053, the Purdue Integrative Data Science Initiative, the Purdue Research Foundation, and the Wabash Heartland Innovation Network. Any opinions, findings and conclusions or recommendations expressed in this material are those of the authors and do not necessarily reflect the views of the sponsors. Further, we would like to thank Mayank Kakodkar for his invaluable feedback and discussion on subgraph function estimation.

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
