[Supplementary Material]

# A The Estimator $\widehat{\Phi}(\mathbf{A}, \boldsymbol{X}; \mathbf{W})$

## A.1 The $k$-CNHON network

**Definition 5** ($k$-CNHON of $G$ given $\mathcal{I}$, or $G^{(k,\mathcal{I})}$). *Let $G^{(k)} = (V^{(k)}, E^{(k)})$ be the higher-order network ($k$-HON) of the input graph $G$, where each node $v^{(k)} \in V^{(k)}$ corresponds to a $k$-node set $C \in \mathcal{C}_{conn}^{(k)}$. For ease of understanding, we will leverage this correspondence and refer to nodes from $V^{(k)}$ with $k$-node sets from $\mathcal{C}_{conn}^{(k)}$ interchangeably. The edge set $E^{(k)}$ is defined such that $E^{(k)} = \{(v_i^{(k)}, v_j^{(k)}) : v_i^{(k)}, v_j^{(k)} \in \mathcal{C}_{conn}^{(k)} \text{ and } |v_i^{(k)} \cap v_j^{(k)}| = k - 1\}$. Moreover, let $\mathcal{I}$ be a set of $k$-nodes sets $\mathcal{I} \subset \mathcal{C}_{conn}^{(k)}$. Then, a $k$-CNHON $G^{(k,\mathcal{I})} = (V^{(k,\mathcal{I})}, E^{(k,\mathcal{I})})$ with supernode $v_{\mathcal{I}}^{(k)}$ is a multigraph with node set $V^{(k,\mathcal{I})} = (V^{(k)} \backslash \mathcal{I}) \cup v_{\mathcal{I}}^{(k)}$ and edge multiset $E^{(k,\mathcal{I})} = E^{(k)} \backslash (E^{(k)} \cap (\mathcal{I} \times \mathcal{I})) \uplus \{(v_{\mathcal{I}}^{(k)}, v^{(k)}) : \exists (u^{(k)}, v^{(k)}) \in E^{(k)}, u^{(k)} \in \mathcal{I} \text{ and } v^{(k)} \notin \mathcal{I}\}$, where $\uplus$ is the multiset union operation.*

## A.2 Proof of Theorem 1

To prove Theorem 1, we assume that $G^{(k,\mathcal{I})}$ has a stationary distribution $\pi$ with

$$\pi(C_i) = \frac{|N^{(k)}(C_i)|}{\sum_{C' \in V^{(k)} \backslash \mathcal{I}} |N^{(k)}(C')| + \sum_{u \in \mathcal{I}} |N^{(k)}(u) \backslash \mathcal{I}|} \ \forall \, C_i \in V^{(k,\mathcal{I})} \backslash \{v_{\mathcal{I}}^{(k)}\},$$

and

$$\pi(v_{\mathcal{I}}^{(k)}) = \frac{\sum_{u \in \mathcal{I}} |N^{(k)}(u) \backslash \mathcal{I}|}{\sum_{C' \in V^{(k)} \backslash \mathcal{I}} |N^{(k)}(C')| + \sum_{u \in \mathcal{I}} |N^{(k)}(u) \backslash \mathcal{I}|}.$$

Fortunately, Wang et al. [65] showed that such a statement is true whenever $\mathcal{I}$ contains at least one $k$-node set from each connected component of $G$ and if each such component contains at least one vertex which is not a part of any $k$-node set in $\mathcal{I}$ and is contained in more than 2 edges in $G$. First, we show that the estimate $\widehat{\Phi}(\mathbf{A}, \boldsymbol{X}; \mathbf{W})$ of each tour is unbiased.

**Lemma 1.** *Let $\mathcal{T}_C^r = (C_i^r)_{i=2}^{t^r}$ be a $k$-node set chain formed by the samples from the $r$-th RWT on $G^{(k,\mathcal{I})}$ starting at the supernode $v_{\mathcal{I}}^{(k)}$. Then, $\forall r \geq 1$,*

$$\mathbb{E}\Big[\sum_{v \in \mathcal{I}} \phi(\mathbf{A}^{(v)}, \boldsymbol{X}^{(v)}; \mathbf{W}) + \Big(\sum_{u \in \mathcal{I}} |N^{(k)}(u) \backslash \mathcal{I}|\Big) \sum_{i=2}^{t^r} \frac{\phi(\mathbf{A}^{(C_i^r)}, \boldsymbol{X}^{(C_i^r)}; \mathbf{W})}{|N^{(k)}(C_i^r)|}\Big] = \Phi(\mathbf{A}, \boldsymbol{X}; \mathbf{W}), \tag{4}$$

*assuming $\Phi(\mathbf{A}, \boldsymbol{X}; \mathbf{W})$ with zero constant.*

*Proof of Lemma 1.* Let's first rewrite Equation (4) as

$$\Big(\sum_{u \in \mathcal{I}} |N^{(k)}(u) \backslash \mathcal{I}|\Big) \mathbb{E}\Big[\sum_{i=2}^{t^r} \frac{\phi(\mathbf{A}^{(C_i^r)}, \boldsymbol{X}^{(C_i^r)}; \mathbf{W})}{|N^{(k)}(C_i^r)|}\Big] = \Phi(\mathbf{A}, \boldsymbol{X}; \mathbf{W}) - \sum_{v \in \mathcal{I}} \phi(\mathbf{A}^{(v)}, \boldsymbol{X}^{(v)}; \mathbf{W}). \tag{5}$$

Since the RWT starts at node $v_{\mathcal{I}}^{(k)}$, we may rewrite the expected value in Equation (5) as

$$\mathbb{E}\left[\sum_{i=2}^{t^r} \frac{\phi(\mathbf{A}^{(C_i^r)}, \boldsymbol{X}^{(C_i^r)}; \mathbf{W})}{|N^{(k)}(C_i^r)|}\right] = \sum_{C_i \in \mathcal{C}_{conn}^{(k)} \backslash \mathcal{I}} \mathbb{E}\left[\mathbb{T}(C_i) \frac{\phi(\mathbf{A}^{(C_i)}, \boldsymbol{X}^{(C_i)}; \mathbf{W})}{|N^{(k)}(C_i)|}\right], \tag{6}$$

where $\mathbb{T}(C)$ represents the number of times the RWT reaches state $C$.

Consider a renewal reward process with inter-renewal time distributed as $t^r$, $r \geq 1$ and reward as $\mathbb{T}(C_i^r)$. Further, note that the chain is positive recurrent, thus $\mathbb{E}[t^r] < \infty$, $\mathbb{E}[\mathbb{T}(C_i^r)] < \infty$ and $\mathbb{T}(C_i^r) < \infty$. Then, from the renewal reward theorem and the ergodic theorem [10] we have

$$\pi(C_i^r) = \mathbb{E}[t^r]^{-1}\mathbb{E}[\mathbb{T}(C_i^r)].$$

Moreover, it follows from Kac's formula [1] that $\mathbb{E}[t^r] = \frac{1}{\pi(v_{\mathcal{I}}^{(k)})}$. Therefore, Equation (6) can be rewritten as

$$\mathbb{E}\left[\sum_{i=2}^{t^r} \frac{\phi(\mathbf{A}^{(C_i^r)}, \boldsymbol{X}^{(C_i^r)}; \mathbf{W})}{|N^{(k)}(C_i^r)|}\right] = \sum_{C_i \in \mathcal{C}_{\mathrm{conn}}^{(k)} \setminus \mathcal{I}} \frac{\pi(C_i)\phi(\mathbf{A}^{(C_i)}, \boldsymbol{X}^{(C_i)}; \mathbf{W})}{\pi(v_{\mathcal{I}}^{(k)})|N^{(k)}(C_i)|}. \tag{7}$$

Now, knowing the stationary distribution of $G^{(k,\mathcal{I})}$, we may simplify Equation (7) to

$$\mathbb{E}\left[\sum_{i=2}^{t^r} \frac{\phi(\mathbf{A}^{(C_i^r)}, \boldsymbol{X}^{(C_i^r)}; \mathbf{W})}{|N^{(k)}(C_i^r)|}\right] = \frac{1}{\sum_{u \in \mathcal{I}} |N^{(k)}(u)\setminus\mathcal{I}|} \sum_{C_i \in \mathcal{C}_{\mathrm{conn}}^{(k)} \setminus \mathcal{I}} \phi(\mathbf{A}^{(C_i)}, \boldsymbol{X}^{(C_i)}; \mathbf{W}), \tag{8}$$

and replace it in Equation (5), concluding our proof. $\qquad\square$

*Proof of Theorem 1.* By Lemma 1, linearity of expectation and knowing that each RWT is independent from the other tours by the Strong Markov Property, Theorem 1 holds. $\qquad\square$

## B  Discussion of MHM-GNN properties

**Conditional independence.**  Although HMNs factorize distributions, the potentials themselves do not provide information on conditional and marginal distributions. Rather, we need to analyze how every pair of variables interacts through all potentials. For the sake of simplicity, consider the model described in Definition 3 for undirected simple graphs, *i.e.* $\boldsymbol{A}_{ij} = \boldsymbol{A}_{ji} \, \forall \, (i,j) \in V^2, \mathbf{A}_{ii} = 0 \, \forall \, i \in V$. If we set $k = 2$, each hyperedge will contain exactly one edge variable and two node variables, which is equivalent to assuming all edges are independent given their nodes' representations. Thus, for $k = 2$ MHM-GNN can recover edge-based models where representations don't use graph-wide information. Furthermore, if we allow the node representation to take graph-wide information, we can recover the recent Graph Neural Networks approaches [21, 29, 8]. If we opt for $k = 3$, a hyperedge defined by nodes $i, j, l$ will contain the set of edge variables $\{\mathbf{A}_{ij}, \mathbf{A}_{il}, \mathbf{A}_{jl}\}$ and node variables $\{\boldsymbol{X}_{i,\cdot}, \boldsymbol{X}_{j,\cdot}, \boldsymbol{X}_{l,\cdot}\}$. Thus, a hyperedge will encompass only edge variables that share one endpoint. In this case, an edge variable $\mathbf{A}_{ij}$ is independent from $\{\mathbf{A}_{lm} : l, m \in V, \{l, m\} \cap \{i, j\} = \emptyset\}$ others given $\{\mathbf{A}_{il} : l \in V\} \cup \{\mathbf{A}_{im} : m \in V\} \cup \{\boldsymbol{X}_{i,\cdot} : i \in V\}$. Thus, MHM-GNN with $k = 3$ can be cast as an instance of the Markov random graphs class proposed by Frank and Strauss [18]. With $k \geq 4$, for every pair of edge variables $\mathbf{A}_{ij}, \mathbf{A}_{lm}$ there exists at least one $C \in \mathcal{C}^{(k)}$ such that $i, j, l, m \subseteq C$. Thus, there exists at least one hyperedge covering every pair of edge variables in the model, resulting in a fully connected hypergraph Markov Network. Therefore, for $k \geq 4$ the model does not assume any conditional independence between edge variables which, since subgraphs share edge variables, is a vital feature for joint $k$-node representations of graphs.

**Exchangeability.**  Although with infinite data and an arbitrary energy function $\phi(\cdot, \cdot; \mathbf{W})$ MHM-GNN would learn a jointly exchangeable [44] distribution, we would like to impose such condition on the model, defining a proper graph model. Equivalently, we would like to guarantee that any two isomorphic graphs have the same probability under MHM-GNN. By definition, the sets of subgraphs from two isomorphic graphs are equivalent under graph isomorphism. Thus, if the subgraph energy function $\phi(\cdot, \cdot; \mathbf{W})$ is jointly exchangeable, the set of subgraph energies from two isomorphic graphs are equivalent. Since the sum operation is permutation invariant and the partition function is a constant, a jointly exchangeable subgraph energy function $\phi(\cdot, \cdot; \mathbf{W})$, such as a GNN, is enough to make MHM-GNN jointly exchangeable.

**Exponential Random Graph Models (ERGMs).** The form of MHM-GNN presented in Definition 3 resembles the general and classical expression of Exponential Random Graph Models (ERGMs) [32]. Indeed, as any energy-based network model, we can cast ours as an ERGM where the sufficient statistics are given by all $k$-size subgraphs. However, we do stress how any exchangeable graph model has a correspondent ERGM representation [33], even when it is not as clear as it in MHM-GNN.

## C  Additional Experiments and Implementation Details from Section 5

### C.1  Results for $k = 5$

Here, we extend the results from Section 5 to a $k = 5$ setting in Table 5 and table 4. Due to the lack of papers with 5 authors (less than 10), we were not able to extend them to the DBLP dataset. Moreover, the conclusions from Section 5 also hold here. That is, MHM-GNN consistently outperforms the baselines. However, on Rent the Runway we see the raw features achieving the highest performance. That is, structural information does not seem to be relevant to this specific task. Nevertheless, we still see that MHM-GNN and GraphSAGE are the methods able to perform the task similarly to the raw features.

Table 4: Balanced accuracy for the **Hyperedge detection** task over subgraphs of size $k = 5$. We report mean and standard deviation over five runs.

| Method | Cora $k = 5$ | Citeseer $k = 5$ | Pubmed $k = 5$ | Steam $k = 5$ | Rent the Runway $k = 5$ |
|---|---|---|---|---|---|
| GS-mean[21] | $0.447 \pm 0.10$ | $0.530 \pm 0.03$ | $0.697 \pm 0.08$ | $0.696 \pm 0.07$ | $0.933 \pm 0.00$ |
| GS-max[21] | $0.384 \pm 0.09$ | $0.543 \pm 0.08$ | $0.722 \pm 0.06$ | $0.765 \pm 0.03$ | $0.940 \pm 0.00$ |
| GS-lstm[21] | $0.422 \pm 0.04$ | $0.525 \pm 0.03$ | $0.736 \pm 0.08$ | $0.532 \pm 0.05$ | $0.557 \pm 0.05$ |
| DGI[64] | $0.504 \pm 0.00$ | $0.500 \pm 0.00$ | $0.500 \pm 0.00$ | $0.626 \pm 0.11$ | $0.827 \pm 0.04$ |
| Raw Features | $0.500 \pm 0.00$ | $0.513 \pm 0.00$ | $0.526 \pm 0.00$ | $0.602 \pm 0.00$ | $\mathbf{0.944} \pm 0.00$ |
| MHM-GNN (Rnd) | $0.460 \pm 0.05$ | $0.453 \pm 0.03$ | $0.493 \pm 0.07$ | $0.748 \pm 0.02$ | $0.924 \pm 0.00$ |
| MHM-GNN | $\mathbf{0.543} \pm 0.06$ | $\mathbf{0.703} \pm 0.04$ | $\mathbf{0.815} \pm 0.10$ | $\mathbf{0.823} \pm 0.00$ | $0.943 \pm 0.01$ |

Table 5: Balanced accuracy for the **DAG Leaf Counting** task over subgraphs of size $k = 5$. We report mean and standard deviation over five runs.

| Method | Cora $k = 5$ | Citeseer $k = 5$ | Pubmed $k = 5$ |
|---|---|---|---|
| GS-mean[21] | $0.223 \pm 0.04$ | $0.259 \pm 0.02$ | $0.284 \pm 0.02$ |
| GS-max[21] | $0.150 \pm 0.07$ | $0.263 \pm 0.01$ | $0.288 \pm 0.02$ |
| GS-lstm[21] | $0.214 \pm 0.03$ | $0.259 \pm 0.00$ | $0.295 \pm 0.04$ |
| DGI[64] | $0.236 \pm 0.02$ | $0.249 \pm 0.00$ | $0.249 \pm 0.00$ |
| Raw Features | $0.251 \pm 0.05$ | $0.266 \pm 0.00$ | $0.290 \pm 0.00$ |
| MHM-GNN (Rnd) | $0.231 \pm 0.01$ | $0.277 \pm 0.02$ | $0.244 \pm 0.01$ |
| MHM-GNN | $\mathbf{0.363} \pm 0.04$ | $\mathbf{0.364} \pm 0.02$ | $\mathbf{0.330} \pm 0.04$ |

### C.2  Hyperparameters and Hyperparameter Search for MHM-GNN

All MHM-GNN models were implemented in PyTorch [46] and PyTorch Geometric [17] with the Adam optimizer [28]. All hyperparameters were chosen to minimize training loss. For learning rate, we searched in {0.01, 0.001, 0.0001} finding the best learning rate to be 0.001 for all models. We used a single hidden layer feedforward network with LeakyReLU activations for both $\rho$ and READOUT functions in all models. Furthermore, following GraphSAGE Hamilton et al. [21], for all models we do an L2 normalization in the motif representation layer, *i.e.* in the output of the READOUT function. Finally, for all models we use $M = 1$ negative example for each positive example. In what follows, we give specific hyperparameters and their search for experiments from Section 5, show results for transductive baselines, and introduce new whole-graph downstream tasks together with their specific hyperparameters and search as well.

### C.3  Pre-trained HMH-GNN for $k$-node downstream tasks (Section 5)

**MHM-GNN architecture.** The energy function of MHM-GNN is as described in Equation (2), where we use a one-hidden layer feedforward network with LeakyReLU activations as $\rho$, a row-wise

sum followed by also a one-hidden layer feedforward network with LeakyReLU activations as the READOUT function and a single layer GraphSAGE-mean Hamilton et al. [21] as the GNN, except for $k = 5$ in the citation networks where we used two layers of the GraphSAGE-mean GNN to achieve faster convergence in training.

**Subsampling positive examples.** We use positive examples subsampled with Forest Fire [35] of size 100 for Cora, Citeseer and DBLP datasets, while for Pubmed, a larger network, we use examples of size 500. For Steam, a smaller network, we use 75 and for Rent the Runway, a mid-size network we use 150.

**Number of tours.** We did 80 tours for all datasets except Pubmed with $k = 4$, which due to a larger $k$-CNHON network, we did 120 tours. A small number of tours will result in high variance in the gradient which, as we observed, tends to impair the learning process. Therefore, we tested training models, each with a different fix number of tours, starting with 1 tour and increasing it 10 by 10 until we reached the reported number of tours, which results in training loss convergence.

**Supernode size.** To construct the supernode, we do a BFS on the $k$-HON of the original input graph, similarly to Teixeira et al. [61]. We have a parameter that controls the maximum number of subgraphs visited by the BFS, which we call supernode budget. This parameter was set to 100K for Pubmed with $k = 3$ and $k = 4$, 5K for Cora with $k = 3$ and $k = 4$, Citeseer with $k = 3$ and DBLP with $k = 3$, 10K for Citeseer with $k = 4$ and 50K for DBLP with $k = 4$. For Steam, we set to 1K for $k = 3$ and to 10K for $k = 4$. For Rent the Runway, we set to 10K for $k = 3$ and to 30K for $k = 4$. For $k = 5$, we used 50K in Cora, 75K in Citeseer, 120K in Pubmed, 50K in Steam and 100K in Rent the Runway. In the same way of tours, we started with a small supernode budget of 100 and increased it by 100 until we observed the tours being completed and the training loss converging.

**Minibatch size.** We used a minibatch size of 50 for Cora, Citeseer and Steam with $k = 3$ and 25 for Cora and Citesser with $k = 4$. For Pubmed, Rent the Runway and DBLP, larger networks, we used minibatches of size 40 for $k = 3$ and 10 for $k = 4$. For Steam, we used 20 for $k = 4$. Again, we tested small minibatch sizes, increasing them until we had training loss convergence and GPU memory space to use. For $k = 5$, we used a minibatch of size 5 in all datasets.

### C.3.1 Transductive baselines

Since we defined the tasks from Section 5 over single graphs in the citation and couathorship networks, in Tables 6a to 6c, and Tables 7a to 7c we show for those datasets results for two prominent transductive node embedding methods, node2vec [19] and DeepWalk [49] together with concatenating the raw features to them, evidencing how even in transductive settings, transductive node embeddings fail to capture joint $k$-node relationships in most settings, performing similarly to the inductive approaches to node representations, thus, performing consistently worse than our MHM-GNN joint $k$-node representations.

### C.4 Pre-trained MHM-GNN representations for whole-graph downstream tasks

In Section 5, we have seen that the motif representations learned by MHM-GNN can better predict hyperedge properties than existing unsupervised GNN representations. In the following experiments we investigate: Are MHM-GNN motif representations capturing graph-wide information (learning $\mathbb{P}(\mathbf{A}, \boldsymbol{X}; \mathbf{W})$)? To this end, inspired by Nair and Hinton [41]'s evaluation of RBM representations through supervised learning, we now investigate if MHM-GNN's pre-trained motif representations can do similarly or better than non-compositional methods that take graph-wide information in (inductive) whole-graph classification.

**Datasets.** We use four multiple graphs datasets, namely PROTEINS, ENZYMES, IMDB-BINARY and IMDB-MULTI [70, 26]. We are interested in evaluating whole-graph representations under two different scenarios, one where the nodes have high-dimensional feature vectors and the other where the nodes do not have features. To this end, we chose the two biological networks PROTEINS and ENZYMES, where nodes contain feature vectors of size 32 and 21 respectively and the social networks IMDB-BINARY and IMDB-MULTI where nodes do not have features. More details in Section D of this supplement.

**Training the model.** Since we have multiple graphs in our datasets, our set of positive graph examples is already given in the data, unlike in Section 5, where we had to subsample positives from

Table 6: Results for transductive baselines in the **Hyperedge Detection** task over $k = 3$, $k = 4$ and $k = 5$ size subgraphs.

| Method | Cora $k = 3$ | Citeseer $k = 3$ | Pubmed $k = 3$ | DBLP $k = 3$ |
|---|---|---|---|---|
| node2vec[19] | $0.534 \pm 0.04$ | $0.525 \pm 0.02$ | $0.501 \pm 0.00$ | $0.461 \pm 0.05$ |
| node2vec[19] + Features | $0.545 \pm 0.01$ | $0.534 \pm 0.01$ | $0.500 \pm 0.00$ | $0.479 \pm 0.04$ |
| DeepWalk[49] | $0.472 \pm 0.02$ | $0.433 \pm 0.01$ | $0.499 \pm 0.00$ | $0.481 \pm 0.00$ |
| DeepWalk[49] + Features | $0.512 \pm 0.01$ | $0.591 \pm 0.01$ | $0.502 \pm 0.00$ | $0.485 \pm 0.02$ |

(a) ($k = 3$) Balanced accuracy for the **Hyperedge Detection** task over subgraphs of size $k = 3$. We report mean and standard deviation over five runs.

| Method | Cora $k = 4$ | Citeseer $k = 4$ | Pubmed $k = 4$ | DBLP $k = 4$ |
|---|---|---|---|---|
| node2vec[19] | $0.537 \pm 0.04$ | $0.513 \pm 0.03$ | $0.504 \pm 0.01$ | $0.405 \pm 0.01$ |
| node2vec[19] + Features | $0.626 \pm 0.03$ | $0.540 \pm 0.01$ | $0.502 \pm 0.00$ | $0.548 \pm 0.10$ |
| DeepWalk[49] | $0.515 \pm 0.07$ | $0.494 \pm 0.10$ | $0.504 \pm 0.01$ | $0.460 \pm 0.01$ |
| DeepWalk[49] + Features | $0.597 \pm 0.05$ | $0.570 \pm 0.01$ | $0.516 \pm 0.01$ | $0.560 \pm 0.03$ |

(b) ($k = 4$) Balanced accuracy for the **Hyperedge Detection** task over subgraphs of size $k = 4$. We report mean and standard deviation over five runs.

| Method | Cora $k = 5$ | Citeseer $k = 5$ | Pubmed $k = 5$ |
|---|---|---|---|
| node2vec[19] | $0.446 \pm 0.08$ | $0.544 \pm 0.08$ | $0.623 \pm 0.13$ |
| node2vec[19] + Features | $0.519 \pm 0.00$ | $0.500 \pm 0.00$ | $0.502 \pm 0.01$ |
| DeepWalk[49] | $0.446 \pm 0.07$ | $0.568 \pm 0.05$ | $0.568 \pm 0.13$ |
| DeepWalk[49] + Features | $0.490 \pm 0.01$ | $0.523 \pm 0.01$ | $0.472 \pm 0.11$ |

(c) ($k = 5$) Balanced accuracy for the **Hyperedge Detection** task over subgraphs of size $k = 5$. We report mean and standard deviation over five runs.

a single graph. The negative examples still need to be sampled. For the biological networks, we used the same negative sampling approach used in Section 5. For the social networks, where the nodes do not have features, for each positive example, we uniformly at random add $n$ edges to it, generating a negative sample (where $n$ is the number of nodes in the graph).

**Experimental setup.** We equally divide the graphs in each dataset between training (unsupervised) and training+testing (supervised). We use two thirds of the graphs in the supervised dataset to train a logistic classifier for the downstream task over the graph's representation. We use a third of the supervised dataset to test the method's accuracy. The classification tasks used here are the same as in Borgwardt et al. [9] and Xu et al. [67]. Again, we set the representation dimension of both MHM-GNN and our baselines to 128. We show results for $k = 3, 4, 5$ motifs representations, $k = n$ whole-graph representations, and unsupervised GNN node representations. To create these representations, we tested both sum and mean pooling for MHM-GNN (except $k = n$) and all the node-based baselines. We report the best performance of each for a fair comparison.

**Baselines.** We compare MHM-GNN against *non-compositional methods*: pooling node representations from GraphSAGE and DGI, directly pooling node features, two recent whole-graph embedding methods, NetLSD [62] and graph2vec [42] and a recent unsuperved whole-graph representation, InfoGraph [58]. Apart from pooling node features, all methods input graph-wide information to their representations. Pooling node features is not applicable to the social networks, since they do not have such information. Additionally, DGI also generates a whole-graph representation to minimize the mutual entropy with the nodes' representations. Note how by setting $k = n$, we consider the entire graph as a single motif and thus, learn a whole-graph representation. Again, all models were trained according to their original implementation.

**Results.** We show in Table 8 the results for whole-graph classification downstream tasks. For each task and each model, we report the mean and the standard deviation of the balanced accuracy (mean recall of each class) achieved by logistic regression over five different runs. We observe how our method consistently outperforms representations computed over the entire graph: the joint DGI approach, graph2vec and node representations pooling. Interestingly, we observe that when the graph has high-dimensional feature vectors of the nodes, pooling small motif representations better

Table 7: Results for transductive baselines in the **DAG Leaf Counting** task over $k = 3$, $k = 4$ and $k = 5$ size subgraphs.

| Method | Cora $k = 3$ | Citeseer $k = 3$ | Pubmed $k = 3$ |
|---|---|---|---|
| node2vec[19] | $0.538 \pm 0.05$ | $0.546 \pm 0.03$ | $0.502 \pm 0.01$ |
| node2vec[19] + Features | $0.556 \pm 0.02$ | $0.527 \pm 0.01$ | $0.501 \pm 0.00$ |
| DeepWalk[49] | $0.466 \pm 0.02$ | $0.503 \pm 0.06$ | $0.499 \pm 0.00$ |
| DeepWalk[49] + Features | $0.543 \pm 0.01$ | $0.584 \pm 0.00$ | $0.503 \pm 0.00$ |

(a) ($k = 3$) Balanced accuracy for the **DAG Leaf Counting** task over subgraphs of size $k = 3$. We report mean and standard deviation over five runs.

| Method | Cora $k = 4$ | Citeseer $k = 4$ | Pubmed $k = 4$ |
|---|---|---|---|
| node2vec[19] | $0.374 \pm 0.06$ | $0.329 \pm 0.04$ | $0.333 \pm 0.00$ |
| node2vec[19] + Features | $0.410 \pm 0.04$ | $0.388 \pm 0.00$ | $0.339 \pm 0.00$ |
| DeepWalk[49] | $0.322 \pm 0.00$ | $0.349 \pm 0.04$ | $0.339 \pm 0.00$ |
| DeepWalk[49] + Features | $0.349 \pm 0.00$ | $0.381 \pm 0.00$ | $0.345 \pm 0.00$ |

(b) ($k = 4$) Balanced Accuracy for the **DAG Leaf Counting** task over subgraphs of size $k = 4$. We report mean and standard deviation over five runs.

| Method | Cora $k = 5$ | Citeseer $k = 5$ | Pubmed $k = 5$ |
|---|---|---|---|
| node2vec[19] | $0.265 \pm 0.05$ | $0.262 \pm 0.03$ | $0.298 \pm 0.03$ |
| node2vec[19] + Features | $0.263 \pm 0.01$ | $0.240 \pm 0.02$ | $0.259 \pm 0.01$ |
| DeepWalk[49] | $0.254 \pm 0.02$ | $0.240 \pm 0.01$ | $0.238 \pm 0.05$ |
| DeepWalk[49] + Features | $0.255 \pm 0.00$ | $0.269 \pm 0.00$ | $0.269 \pm 0.01$ |

(c) ($k = 5$) Balanced Accuracy for the **DAG Leaf Counting** task over subgraphs of size $k = 5$. We report mean and standard deviation over five runs.

Table 8: Results for the whole-graph classification task evaluated over balanced accuracy. We report mean and standrad deviation over five runs.

| Method | PROTEINS | ENZYMES | IMDB-BIN. | IMDB-MULT |
|---|---|---|---|---|
| GS-mean[21] | $0.753 \pm 0.01$ | $0.435 \pm 0.02$ | $0.454 \pm 0.01$ | $0.347 \pm 0.01$ |
| GS-max[21] | $0.729 \pm 0.01$ | $0.400 \pm 0.04$ | $0.447 \pm 0.01$ | $0.360 \pm 0.01$ |
| GS-lstm[21] | $0.739 \pm 0.01$ | $0.404 \pm 0.04$ | $0.442 \pm 0.00$ | $0.342 \pm 0.01$ |
| DGI (Nodes)[64] | $0.743 \pm 0.02$ | $0.349 \pm 0.04$ | $0.469 \pm 0.00$ | $0.367 \pm 0.02$ |
| DGI (Joint)[64] | $0.756 \pm 0.00$ | $0.263 \pm 0.03$ | $0.568 \pm 0.03$ | $0.376 \pm 0.01$ |
| Raw Features | $0.665 \pm 0.05$ | $0.210 \pm 0.02$ | $-$ | $-$ |
| NetLSD[62] | $0.760 \pm 0.00$ | $0.250 \pm 0.00$ | $0.550 \pm 0.00$ | $0.430 \pm 0.01$ |
| graph2vec[42] | $0.685 \pm 0.00$ | $0.166 \pm 0.00$ | $0.507 \pm 0.00$ | $0.335 \pm 0.00$ |
| InfoGraph[58] | $0.690 \pm 0.04$ | $0.278 \pm 0.04$ | $\mathbf{0.691} \pm 0.04$ | $\mathbf{0.466} \pm 0.02$ |
| MHM-GNN (Rnd) ($k = 3$) | $0.733 \pm 0.01$ | $0.293 \pm 0.02$ | $0.586 \pm 0.00$ | $0.369 \pm 0.001$ |
| MHM-GNN ($k = 3$) | $\mathbf{0.777} \pm 0.01$ | $\mathbf{0.445} \pm 0.01$ | $0.586 \pm 0.00$ | $0.376 \pm 0.00$ |
| MHM-GNN (Rnd) ($k=4$) | $0.720 \pm 0.02$ | $0.229 \pm 0.04$ | $0.580 \pm 0.00$ | $0.371 \pm 0.00$ |
| MHM-GNN ($k = 4$) | $\mathbf{0.780} \pm 0.02$ | $0.390 \pm 0.04$ | $0.621 \pm 0.00$ | $0.390 \pm 0.002$ |
| MHM-GNN (Rnd) ($k = 5$) | $0.722 \pm 0.01$ | $0.213 \pm 0.03$ | $0.580 \pm 0.00$ | $0.378 \pm 0.005$ |
| MHM-GNN ($k = 5$) | $\mathbf{0.773} \pm 0.01$ | $0.326 \pm 0.04$ | $0.600 \pm 0.01$ | $0.397 \pm 0.001$ |
| MHM-GNN (Rnd) ($k = n$) | $0.704 \pm 0.03$ | $0.266 \pm 0.02$ | $\mathbf{0.707} \pm 0.02$ | $\mathbf{0.446} \pm 0.005$ |
| MHM-GNN ($k = n$) | $0.753 \pm 0.00$ | $0.327 \pm 0.01$ | $\mathbf{0.694} \pm 0.02$ | $\mathbf{0.451} \pm 0.01$ |

generalizes than all other methods to unseen graphs. On the other hand, we observe that using a joint whole-graph representation, either with $k = n$ in our model or with NetLSD or with InfoGRAPH, can perform better without node features. In fact, there is no significant difference between using a random and a trained model for the joint representation. It is known how a random GNN model simply assigns a unique representation to each class of graphs indistinguishable under the 1-WL test [67]. Therefore, for graphs without node features, assigning unique representations seems to be the best in this setting, which means that the tested graph embedding and unsupervised representation methods are not really capturing significant graph information. Overall, we observe that indeed motif representations are capable of representing the entire graph to which they belong and even give better results, evidencing how MHM-GNN is learning graph-wide information, *i.e.* capturing $\mathbb{P}(\mathbf{A}, \mathbf{X}; \mathbf{W})$ and how motif compositionality can explain networks functionality.

**MHM-GNN architecture.** We use the same $\rho$ and READOUT functions as in Section 5, while changing the GNN to GIN Xu et al. [67] (which gave better validation results than the GAT, GCN,

and GraphSAGE GNNs). Again, we use $M = 1$, *i.e.*, we sample one negative example for each positive sample. We show results of MHM-GNN for $k = 3, 4, 5, n$. For the estimator $\widehat{\Phi}(\mathbf{A}, \boldsymbol{X}; \mathbf{W})$, we perform 30 tours for every model and dataset.

**GNN layer.** We use a single-layer GIN Xu et al. [67] as the GNN layer in our method. For $k = n$, where the GNN is applied over large graphs, we used GIN with two layers. Note that we also tested GraphSAGE-mean, GCN and GAT GNN layers here, but GIN resulted in faster training loss convergence.

**Number of tours.** We did 30 tours for all datasets. Again, we tested training models, each with a different fix number of tours, starting with 1 tour and increasing 10 by 10 until we reached the reported number of tours, which results in training loss convergence.

**Supernode size.** We did a BFS with the maximum number of subgraphs visited as 5K for all models (and all $k$). Again, we started with a small supernode budget of 100 and increased it by 100 until we observed the tours being completed and the training loss converging.

**Minibatch size.** We used a minibatch size of 50 for ENZYMES and PROTEINS for all reported $k$. For IMDB-BINARY and IMDB-MULTI, which have larger networks we used a minibatch size of 10. Again, we tested small minibatch sizes and increased until we had training loss convergence and GPU memory to use.

**Pooling functions.** We tested both sum and mean pooling motif (our model) and node (baselines) representations for all models here. We observed that mean pooling performs the best for all models in all datasets, except for the ENZYMES dataset, where sum pooling performed the best for all models. Thus, Table 8 contain results with mean pooling for all models in the PROTEINS, IMDB-BINANRY and IMDB-MULTI datasets and sum pooling for all models in the ENZYMES dataset.

## D  Datasets

We present the datasets statistics in Table 9 and Table 10. For the PROTEINS and ENZYMES datasets, we added the node labels as part of the node features. For the DBLP, we subsampled (with Forest Fire) the original large network from Yadati et al. [69]. For the Steam graphs, we consider user-product relations from 2014 to create the training graph and data from 2015 to create the test graph. Similarly, we use 2016 data to create the Rent the Runway training graph and 2017 data to create the test graph. For both product networks, the node features we created are sparse bag-of-words from the user text reviews.

Table 9: Single graph datasets statistics.

| Dataset | Type | Nodes | Edges | Features |
|---|---|---|---|---|
| Cora [55] | Citation Network | 2,708 | 5,429 | 1,433 |
| Citeseer [55] | Citation Network | 3,327 | 4,732 | 3,703 |
| Pubmed [55] | Citation Network | 19,717 | 44,338 | 500 |
| DBLP [69] | Coauthorship Network | 4,309 | 12,863 | 1,425 |
| Steam [47] (Train) | Product Network | 1,098 | 7,839 | 775 |
| Steam [47] (Test) | Product Network | 1,322 | 7,547 | 775 |
| Rent the Runway [38] (Train) | Product Network | 2,985 | 55,979 | 1,475 |
| Rent the Runway [38] (Test) | Product Network | 5,003 | 67,365 | 1,475 |

Table 10: Multiple graphs datasets statistics.

| Dataset | Type | Graphs | Features | Classes |
|---|---|---|---|---|
| PROTEINS [26] | Biological Network | 1,113 | 32 | 2 |
| ENZYMES [26] | Biological Network | 600 | 21 | 6 |
| IMDB-BINARY [26] | Social Network | 1,000 | 0 | 2 |
| IMDB-MULTI [26] | Social Network | 1,500 | 0 | 3 |

# E    Related Work: Higher-order Graph Representations

In what follows, we review the existing approaches to higher-order graph representations in literature.

**Higher-order graph representations.** Morris *et. al* [39] showed how to expand the concept of a GNN, an approach based on the 1-WL algorithm [66], to a $k$-GNN, an approach based on the class of $k$-WL [12] algorithms, where instead of generating node representations, one can derive higher-order ($k$-size) representations later used to represent the entire graph. Although such approaches to represent entire graphs have been recently used in *supervised* graph classification tasks, how to systematically use them in an inductive unsupervised manner was not clear. Since edge-based models require factorizing over a 2-node representation, only 1-WL [30, 67, 21, 63] and 2-WL [39]-based GNNs can be used. Additionally, $k$-GNNs can be thought of as a GNN over an extended graph, where nodes are $k$-node tuples and edges exist between $k$-tuples that share exactly $k - 1$ nodes. One could indeed think of applying an edge-based loss to the extended graph, where the nodes ($k$-node tuples) representations are given by a $k$-GNN. However, an edge-based model assumes independence among edges and an edge in the extended graph is repeated several times in the extended graphs, thus they are not independent. Finally, even if one could provide an unsupervised objective to $k$-GNNs, it would still require $\mathcal{O}(n^k(k\delta)L)$ steps to compute an $L$-layer $k$-GNN over a graph with $n$ nodes and maximum degree $\delta$. Due to the non-linearities in the READOUT function and in the neighborhood aggregations in $k$-GNNs, unbiased subgraph estimators such as the one presented in this work and neighborhood sampling technique such as the one from Hamilton et al. [21] would not provide an unbiased or a bounded loss estimation such as MHM-GNN does. Moreover, the more recent sparser version of $k$-GNNs [13] uses $k$-node tuple representations, instead of $k$-node subgraph represenations as in the original paper. Finally, MHM-GNN can take advantage of any graph representation method, including $k$-GNNs [39] and non-GNN approaches such the ones presented in Relational Pooling [40].

**Sum-based subgraph representations.** There has been recent work representing subgraphs by equating them with sets of node representations [22]. In general, these approaches use graph models able to generate node representations and then add a module on top to aggregate these individual representations in the downstream task. The most prominent efforts have treated subgraph representations as sums of the individual nodes' representations [22], namely sum-based techniques. These approaches do not rely on joint subgraph representations, *i.e.* subgraphs that share nodes will tend to have similar representations, constraining their representational power and thus relying more on the downstream task model.

**Hypergraph models.** In this work, we wish to learn a graph model through motif representations in the presence of standard dyadic (graph) data, *i.e.* we are only observing pairwise relationships. Therefore, we emphasize that hypergraph models, despite dealing with higher-order representations of graphs, require observing polyadic (hypergraph) data and therefore are not an alternative to the problem studied here.

**Supervised learning with subgraphs.** Meng et al. [37] made the first effort towards supervised learning with subgraphs, where the authors predict higher-order properties from temporal dyadic data, as opposed to the problem presented here, where we are are interested in *inductive unsupervised* learning of $k$-node sets from static graphs. Moreover, Meng *et. al* learned subgraph properties while optimizing a pseudo-likelihood function, *i.e.* ignoring the dependencies among different subgraphs in the loss function. Because different node sets share edge variables, it is vital to learn dependencies among them. Hence, here we presented the first graph model based on $k$-size motif structures trained with a proper Noise-Contrastive Estimation function, *i.e.* our model accounts for dependencies between every edge to represent $k$-size node sets.