[Reviews · NeurIPS 2020]

Review 1

Summary and Contributions: This paper introduces MHM-GNNs an inductive unsupervised learning approach to learn k-node joint representations of graph structured data. The proposed model is theoretically interesting but in its naive form suffers from high computational cost, thus making it difficult to scale. To address this limitation the authors also propose an unbiased MCMC sampling scheme based on Random Walk Tours CIS subgraphs which allows them to bound the overall NCE objective. Empirically, the proposed approach yields better representations for downstream classifiers for hyperedge and DAG leaf counting tasks when compared to other baselines.

Strengths: The proposed approach to use MHM GNNs to learn k-node representations is a very interesting idea which has potential application to multiple domains as mentioned. The authors do a good job alluding to the inherit difficulties in scaling this approach naively and as such gradient estimation of the partition function in this setting becomes increasingly important. As far as I know, this represents the first unsupervised learning approach to learn k-node representations and thus is inherently interesting given the burgeoning literature on higher-order GNN's and the limitations of current GNN's.

Weaknesses: While the problem setting and proposed approach are interesting there are some drawbacks in the execution of this idea. First much of the experimental detail is left to the supplementary material and makes the main paper appear lacking in results. Concerningly, few of the transductive baselines outperform the main baselines (see Cora table in the Appendix for Deepwalk + features) reported in the main body of the paper and thus their omission is questionable. Furthermore, the chosen datasets as the paper recognizes are either small graphs or contain only a single graph and as a result its difficult to assess how scalable the proposed approach is to larger real world graphs. The biggest weakness in this reviewers opinion is that its unclear why the MCMC scheme proposed is a natural or superior choice to existing approaches to training EBMs in the literature. Training EBMs have seen a resurgence of late and there have been multiple approaches that provide significant computational benefit [1] [2] [3] are few recent examples. This is without considering classical approaches such as Contrastive Divergence, Persistent-CD, Langevin dynamics to sample negatives, etc ... Granted, the partition function in this paper has a combinatorial component but without experiments its unclear if existing methods help. Thus I believe it is critical that the authors either convincingly argue why other approaches to training EBMs are unsuited here or demonstrate empirically that they fall short in this setting. Finally, turning to the experiments reporting balanced accuracy while not technically incorrect is not a common choice for these datasets (albeit the task is different). More common metrics include AUC/AP but this is a minor concern. What is perhaps more awkward is how the random feature baseline outperforms many GS variants, suggesting that any structural information is detrimental to the task of predicting hyperedges. This is rather alarming as one would expect a small non-negligible benefit not a detriment which suggests that there are issues during training the GS and perhaps hyperparams could be more carefully tuned. I am willing to change my score if the authors can suitably rebut my points. [1] On the Anatomy of MCMC-Based Maximum Likelihood Learning of Energy-Based Models by Nijkamp et. al [2] Generative Modeling by Estimating Gradients of the Data Distribution by Song and Ermon [3] Flow Contrastive Estimation of Energy-Based Models by Gao et. al

Correctness: I did not go through the proofs in the supplement in detail but one thing that was unclear to me is the correctness of Eqn 3. I believe this shouldn't be an equality but an inequality due to the constant. But this could be my misunderstanding. Also, it was unclear from the paper why SGD would take O(n^k) operations, this is not obvious to me.

Clarity: The paper in this reviewers opinion suffers from a lack of clarity in writing with an excess of notational jargon. I would urge the authors to rewrite portions of the draft with an eye towards readibility rather than obscure definitions. Take for instance the description of k-HON and k-CHON, the writing here is difficult to parse and instead a small illustrative figure would be extremely helpful. Consequently, the statement of Theorem 1 also becomes difficult to parse even though I do not have reasons to doubt its correctness. Furthermore, the presentation could benefit from a clearer exposition of the Experimental setting. Much of this detail already exists in the appendix ---i.e. dataset statistics, but they should be included in the main paper. In this reviewers opinion the paper relies too heavily on details that are left to the appendix and some datasets are entirely reserved for the appendix. Much of this could be alleviated by reducing mathematical jargon, for example things like CIS this could be stated upfront in the beginning of the method.

Relation to Prior Work: This paper could benefit from a detailed related work section on training EBMs. There is a large body of work that is extremely relevant and its unclear why its not discussed as many of the techniques could prove to be relevant baselines or approaches that could immediately benefit MHM-GNN's.

Reproducibility: Yes

Additional Feedback: Post-Rebuttal: I thank the authors for their rebuttal and I have increased my score to 6. After discussions with other reviewers I feel this paper has interesting ideas that could be of value to both the GRL and EBM communities. I will also retract my previous criticism on the random baseline performing well on the k-ary tasks compared to vanilla GNN's as any k-ary task is difficult for current GNN's. I encourage the authors to address the presentation issues as well as including a disscusion on related work for EBM's. I would also encourage the authors to consider adding a k-GNN baseline as asked by other reviewers.


Review 2

Summary and Contributions: The authors propose the task of unsupervised representation learning of k-ary tuples of nodes within an input graph, with direct application to various k-hyperedge-detection tasks. The approach realising this task is the MHM-GNN, which explicitly learns these representations via a combination of a GNN and permutation-invariant functions over induced k-tuple subgraphs, coupled with an energy-based model over these tuple representations. The EBM is then trained using a combination of a contrastive loss and a optimisation trick based on random-walks. Results clearly illustrate that such encoders learn representations that are meaningful to hyperedge discovery tasks.

Strengths: - A clear and well-motivated problem statement which was previously largely overlooked. Obvious application to detecting k-product carts. - The application of EBMs and proposed tricks seems well-motivated to retaining scalability of the proposed method, especially compared with prior work on motif-based representation learning such as k-GNNs. - The results seem strong and clearly indicate the outperformance of the method.

Weaknesses: Main limitation in my opinion is---even if trivial---a lack of comparison to the thoroughly-cited work on higher-order GNNs and similar, or a lack of explanation why such a comparison wasn't performed. The narrative of the paper makes me believe that models like k-GNNs could be a viable baseline, so I was surprised to not find it (even in the Appendix). The authors also may find it useful to consider a recently-published scalable extension to k-GNNs (by Morris et al, at GRL+ ICML'20 Workshop): https://grlplus.github.io/papers/80.pdf Further, for the graph-level prediction tasks presented in the Appendix, I would like to draw the authors' attention to InfoGraph (Sun et al., ICLR'20). InfoGraph is a specialised extension of DGI, which makes specific modifications to the pipeline that are tuned for graph-level tasks, and a comparison against InfoGraph (+ appropriate citation) would be a good idea. I would consider improving my score if the comparisons outlined above (or explanation for lack thereof) are included in the rebuttal.

Correctness: For the most part, the claims of the paper seem to be correct. I will note that I'm not an expert in the particular subfield utilised here so I cannot guarantee the correctness of the theory, but the proofs seemed okay to me. I have a few thoughts about the attribution of existing work (DGI) that I would like the authors to address. Specifically, regarding the following two passages: “If we want to tackle downstream tasks that require jointly reasoning about k > 2 nodes, but whose input data are dyadic relations (i.e., standard graphs) rather than hyperedges, we must develop techniques that can go beyond edge losses.” and “significant gap between edge and supervised whole-graph losses (i.e., we need multiple labeled graphs for these to work)” How would the existing work on DGI (and similar approaches) map into these limitations? Given that DGI didn't use a purely edge-wise loss but rather a local-global objective that considered graph level readouts. A comment on this in the paper would be useful.

Clarity: The paper is well-written, and there are really no issues here. I would just like to point out a possible mis-attributed citation: “are equatable to other more powerful node representations, such as those in Hamilton et al. [13], Veličković et al. [40]”. I think the citation of Veličković et al. should be GAT rather than DGI in this context? This is because DGI doesn't really present a GNN architecture (it re-uses the GCN).

Relation to Prior Work: The paper does a great job casting its contributions within the existing related (and unrelated) work on graph representation learning. - As mentioned above, I would recommend including a citation of InfoGraph, as a recent relevant extension to DGI. - Further, the authors may wish to cite Graph Markov Neural Nets (GMNN; Qu and Tang, ICML'19), which also directly combine GNNs with concepts from probabilistic graphical models.

Reproducibility: Yes

Additional Feedback: I've read the other reviews as well as the authors' rebuttal. Thank you for your efforts! The InfoGraph experiments are valuable. I'll retain my score (weak accept), given that I am still satisfied with the paper's direction and findings. However, I remain unconvinced by the authors' remark about not including the k-GNN baselines. In my opinion, even if they promote a somewhat incorrect objective, k-GNNs should be included at least in the Appendix (+ a clear disclaimer on why this would be a bad thing to do). Current explanation provided in the main paper is not sufficiently convincing.


Review 3

Summary and Contributions: They paper presents a representation learning method for k-node subgraphs of a larger graph. This method advances upon prior work which learns representations for single nodes or edges in a graph. The method is based on learning a generative model over graphs. This generative model takes the form of an energy-based model whose energy function consists of a sum of energies of all k-node subgraphs in in the graph (product-of-experts). Each expert's energy is given from a graph neural net acting on a k-node subgraph. Given the model is unnormalized, it is trained with NCE. This is still challenging as the NCE objective requires computing the energy which consists of a sum over an exponentially large set of subgraphs. The authors present an unbiased estimator for the energy which results in an upper bound on the original NCE objective which can be tractable optimized. This unbiased estimator is based on the intuition that most k-node subgraphs will be unconnected (and therefore have identical energy) and an unbiased sampling algorithm which can quickly sample non-empty subgraphs. The authors apply this representation learning method to various graph datasets and utilize their learned representation for downstream prediction of the properties of subgraphs. They demonstrate that their approach has superior performance when compared to previous methods which only learn single node or edge representations.

Strengths: I enjoyed this paper. I think it is a compelling application of EBMs, NCE, and MCMC. This is an important problem and the proposed method seems like solid progress upon prior work (although I am no expert in the graph representation learning space). Particularly, I enjoy how the energy-based parameterization of the likelihood easily allows for the learning problem to be based on permutation invariant generative model of the data which makes far fewer simplifying assumptions than single node or edge models of graphs. The paper is clearly written and the method appears to be sound. The experimental results are impressive and make a notable improvement over the baseline methods of which there are many shown for comparison.

Weaknesses: I found figure 1 to be somewhat confusing to follow, especially given its placement I found it uninformative. The model as defined in section 3 was clear but the figure does little to reinforce that. I think it could be made more clear if the selection of the k-node subgraphs was also visualized in some way and then visualized how they are composed to generate the whole-graph energy. Along those lines, I think that the notation developed in section 3 could be reinforced with some figures as well showing pictorially what each symbol represented. The least clear part of the paper was the explanation of the subgraph sampling procedure in section 4. This could also be greatly reinforced with some figures showing how one maps a standard graph to a k-HON and a k-CNHON.

Correctness: The claims and methods appear correct to me.

Clarity: The paper is clearly written. Its motivation if well stated and the model is well defined. As written in the weaknesses section, some of the more graph-heavy notation could be reinforced with some figures which would make the paper much more accessible to people like myself who are not very familiar with graph-based machine learning.

Relation to Prior Work: The authors clearly state related work in the graph setting to my satisfaction. There is much recent work on EBMs which was not included. Little of it is related to graphs but some references to other recent EBM work would be desirable. Similarly, there is much recent work on contrastive objectives for representation learning which are related but not included.

Reproducibility: Yes

Additional Feedback: I enjoyed this work. I think it is a good paper but could be greatly improved in the ways I have sated above. -------------------------------- After rebuttal and discussion ------------------------------- I enjoyed this work and feel it is very interesting from the perspective of EBMs. The discussion brought many issues to my attention from the graph-modelling side of things. These insights do not change my view of this work but do make me feel less confident in my judgement. I will keep my score the same but reduce my confidence.


Review 4

Summary and Contributions: This paper proposed a novel unsupervised inductive model which combines GNNs and hypergraph Markov networks to learn joint k-node representations (motif representations). The authors further proposed an unbiased MCMC estimator to efficiently estimate the total energy of the graph to solve the computational instability of directly optimizing the loss function w.r.t. the C(n, k) induced subgraphs of a graph. Experiments on six datasets and two downstream tasks demonstrated the effectiveness of the proposed model.

Strengths: The authors proposed a GNN based model to learn motif (of a fixed size k) representations in an unsupervised manner. The learnt representations can then be used in many downstream tasks. The experiments demonstrated the superiority of the proposed model. The method is a general framework for k-node representation learning and can be used in a variety of applications.

Weaknesses: - To effectively compute the total energy of the C(n, k) subgraphs of a graph, the authors propose to approximate the total energy using only the energy of connected induced subgraphs (CISes) in Eq. (3), while assuming the energy of disconnect ones is constant. The authors could give some theoretical bounds or empirical analysis to show how well does this approximation holds. - The authors mentioned that when k=2, MHM-GNN actually recovers existing edge-based models, such as GraphSAGE. I think the authors should conduct an experiment to compare the proposed MHM-GNN to the existing edge-based models using k=2. If the model is comparable to the edge-based ones in such scenario, the generality of the model can be demonstrated. - I am curious why simply summing the raw node features outperforms GraphSAGE and DGI in many cases, observing from Table 1. The authors should provide an explanation to this phenomena. - The nature of hypergraph Markov networks inherently allows us to define a potential over a set of nodes of arbitrary size k. In many real-world applications, the size of the motif is not fixed, like the co-authorship problem used in the hyperedge detection task. However, in the proposed MHM-GNN model, k is fixed. I am curious how the model can be adapted to allow flexible k, as the overall framework permits it. - The authors only use a READOUT function of a row-wise sum followed by a multi-layer perceptron to pool the subgraph features. The authors could explore a variety of READOUT functions like permutation invariant LSTM to compare their performance. - I am also interested to see when supervised information is available such as the information used in the hyperedge detection task, how can we integrate this information into representation learning?

Correctness: yes

Clarity: yes

Relation to Prior Work: yes

Reproducibility: Yes

Additional Feedback:

[Author Response · NeurIPS 2020]

Thank you all for all the positive comments and useful suggestions. We address each reviewer separately, breaking down into topics when possible. References are at the end.

**Reviewer #1**

**Misunderstanding of experiments.**

1. Our approach is **inductive**, while the baselines in the appendix are transductive. **Transductive** methods **are supposed to be better than inductive methods on transductive tasks**, since the transductive methods know that the examples in train and test data are the same. **To our great surprise, our approach is better in 13 out of 14 experiments** over transductive methods. This showcases the strength of our inductive method. Nevertheless, these positive results belong to the appendix since our work focuses on inductive tasks.

2. Recent theoretical results [2] (and older empirical ones [1]) show that GNN node representations cannot perform $k$-ary tasks. **The poor performance of GNNs that output node representations was no surprise.**

**Misunderstanding of theory.**

Line 37 in our paper shows that the challenge is overcoming the **intractability** of the **unormalized** probability function $\Phi(\mathbf{A}, \mathbf{X}; \mathbf{W})$, which is a challenge specific to $k$-ary tasks on graphs. Simply, there is no existing EBM method that deals with this issue. The "intractability" of the normalization factor $Z$ is easy to solve using standard methods (NCE). We will further clarify this point in our paper.

**Reviewer #2.** Great suggestions!

**New InfoGraph results.**

Thank you for the great reference, we were not aware of InfoGraph! **We ran InfoGraph experiments** and found it comparable to our approach (with $k = n$) on IMDB tasks but significantly worse than our approach (with $k = 3, 4, 5$) on ENZYMES and PROTEINS (just for reference, the new results are: PROTEINS: $0.690 \pm 0.04$ ENZYMES: $0.278 \pm 0.04$ IMDB-BIN: $0.691 \pm 0.04$ IMDB-MULT: $0.466 \pm 0.02$). These do not affect our conclusions and we will add these results to the final version of the paper.

**Why not $k$-GNNs.**

The main paper (line 79) discusses why $k$-GNNs should not be used in our tasks (more at line 569 (appendix)). In short, adapting $k$-GNNs to our task would mean proposing an incorrect objective function for optimization, since each edge would appear $\binom{n-2}{k-1}$ times in the objective function with different (inconsistent) predictions for each possible subgraph. We fear this could be misconstrued as our endorsement for this optimization objective.

**DGI and edge-base loss.**

The main paper (line 72) discusses why DGI is not an edge-based model. Recent work [2] shows how single-node GNN representations are theoretically incapable of jointly representing $k$ nodes, thus DGI cannot be trivially extended to joint $k$-node representations.

**Reviewer #3**. Thanks for the great suggestions!

We will add to the appendix a figure with a graph, its $k$-HON and its $k$-CNHON with an example tour. We will also move Figure 1 to Section 3 and incorporate your suggestions. Furthermore, we will add references to works on EBMs in the related work of our supplement, making it clear that existing work do not handle an intractable *unormalized energy* of our type.

**Reviewer #4**. Thanks for the great comments!

**Connected Induced Subgraph (CIS).**

There is a large body of empirical work in the complex networks literature evidencing how CIS'es are effective in representing graphs, for example see [3]. We will add those references as motivation for using CIS'es in section 4.

**Why node-based GNNs underperform in $k$-ary tasks.**

Recent theoretical results [2] (and older empirical ones [1]) show that GNN node representations cannot perform $k$-ary tasks. **The poor performance of GNNs that output node representations was no surprise.** The simple use of raw features performs better. We will emphasize the theory in the results section.

**Future work.**

The reviewer proposes challenging but exciting future work. Making $k$ flexible by simply decomposing the energy into a sum of different $k$'s does not guarantee that the model will not overfit on some specific $k$ value, not learning with the other subgraph sizes. Coping with these challenges is an exciting area for future work.

**References:**
[1] Meng et al. "Subgraph pattern neural networks for high-order graph evolution prediction." AAAI. 2018.
[2] Srinivasan & Ribeiro. "On the Equivalence between Positional Node Embeddings and Structural Graph Representations." ICLR 2020.
[3] Milo et al. "Network motifs: simple building blocks of complex networks." Science 2002.


[Meta-Review · NeurIPS 2020]

This paper generated a lot of discussion and ultimately all referees agreed that the main ideas of the paper are interesting and well-motivated, while providing new directions for the graph learning literature. There are a few concerns on presentation issues, but these can be addressed in a camera version. For these reasons, the consensus is to recommend accepting this paper. The paper makes extensive use of k-ary prediction tasks but ignores a large body of literature considering that topic. Thus, for the camera version, the authors are encouraged to consider the following papers and references therein, and augment the related work and baselines as appropriate: -- Zhang et al. Beyond link prediction: Predicting hyperlinks in adjacency space. AAAI 2018. -- Benson, Austin R. et al. Simplicial closure and higher-order link prediction. PNAS, 2018. -- Xu et al. "Hyperlink prediction in hypernetworks using latent social features" International Conference on Discovery Science, 2013. -- Zhang et al. Recovering metabolic networks using a novel hyperlink prediction method, arXiv:1610.06941 (2016). -- Patania et al. The shape of collaborations. EPJ Data Science 2017. -- Patil et al. Negative Sampling for Hyperlink Prediction in Networks. PAKDD 2020 -- Yoon et al. How Much and When Do We Need Higher-order Information in Hypergraphs? A Case Study on Hyperedge Prediction. WWW 2020.